# CARD: Towards Conditional Design of Multi-Agent Topological Structures

**Tongtong Wu**[1,*,†], **Yanming Li**[2,*], **Ziye Tang**[2], **Chen Jiang**[2], **Linhao Luo**[1], **Guilin Qi**[2],
**Shirui Pan**[3], **Gholamreza Haffari**[1]
[1]Monash University [2]Southeast University [3]Griffith University

## Abstract

Large language model (LLM)-based multi-agent systems have shown strong capabilities in tasks such as code generation and collaborative reasoning. However, the effectiveness and robustness of these systems critically depend on their communication topology, which is often fixed or statically learned, ignoring real-world dynamics such as model upgrades, tool changes, or knowledge source variability. To address this limitation, we propose CARD (Conditional Agentic Graph Designer), a conditional graph-generation framework that instantiates AMACP, a protocol for adaptive multi-agent communication. CARD explicitly incorporates dynamic environmental signals into graph construction, enabling topology adaptation at both training and runtime. Through a conditional variational graph encoder and environment-aware optimization, CARD produces communication structures that are both effective and resilient to shifts in model capability or resource availability. Empirical results on HumanEval, MATH, and MMLU demonstrate that CARD consistently outperforms static and prompt-based baselines, achieving higher accuracy and robustness across diverse conditions. The source code is available at:
`https://github.com/Warma10032/CARD`.

## 1 Introduction

Multi-agent systems (Liu et al., 2025; Li et al., 2026) powered by large language models (LLMs) (OpenAI, 2024a; Chen, 2024) have recently demonstrated remarkable capabilities across a wide range of complex tasks, from code synthesis (Chen et al., 2023) to collaborative reasoning (Liu et al., 2023). By integrating each model's internal knowledge, natural language generation, and inference abilities with external tools (Zhang et al., 2024a;b) or peer LLMs, these systems effectively decompose problems (Yao et al., 2023), coordinate subgoals (Liang et al., 2024), and integrate diverse information sources (Lee et al., 2023; Min et al., 2025). However, the communication topology, which specifies how agents are interconnected, significantly influences performance, affecting both solution quality and robustness to evolving conditions such as model upgrades (Wu et al., 2024a), API modifications (Wu et al., 2026; 2024b), and fluctuating data sources (Min et al., 2025).

Current topology design approaches typically fall into two categories (Bei et al., 2025; Liu et al., 2025). Many systems depend on manually crafted pipelines (Hong et al., 2024) or predefined agent sequences (Wu et al., 2023), which perform effectively in stable, well-understood scenarios but lack adaptability. Conversely, recent methods automatically learn communication structures by backpropagating through "text gradients" (Zhuge et al., 2024) or parameterizing inter-agent connections via differentiable modules (Zhang et al., 2025a). Yet, these learned topologies generally assume static environments, failing to account for transient external factors. Consequently, when conditions change, such as upgrading a model base (e.g. GPT-4o $\rightarrow$ GPT-5), tool availability variations, or deterioration in data source quality, static or naively learned topologies become fragile, resulting in redundant interactions or disrupted information flows (see Figure 1).

---

[1,*] These authors contributed equally to this work.
[2,†] Corresponding author: tongtong.wu@monash.edu.

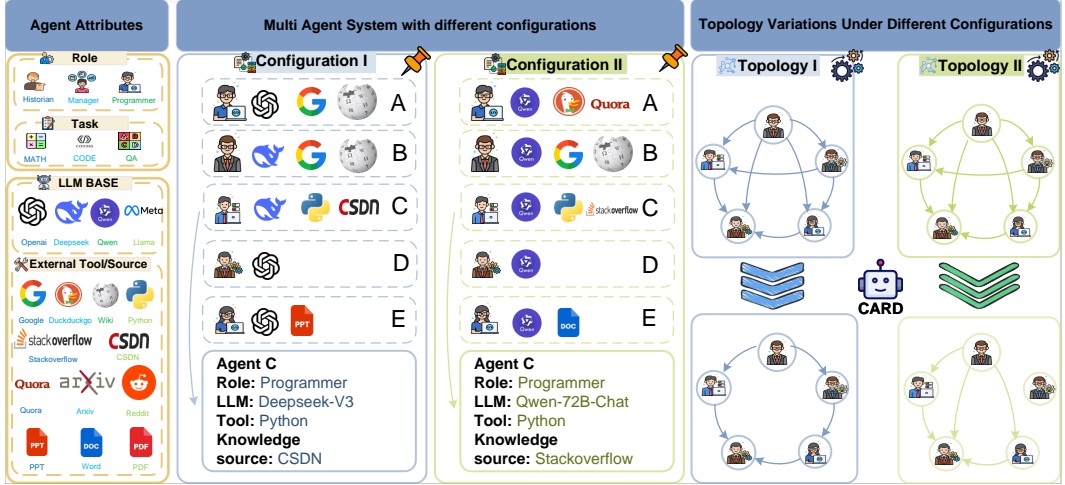

Figure 1: Agent attributes and corresponding communication topologies under two environmental configurations, illustrating that topology is determined by both task requirements and the capabilities of the model base and available resources.

---

**Adaptive LLM-based Multi-agent Communication Protocol (AMACP):**

Given a task/query $q$, an optimal communication topology for $q$ should satisfy the following protocol logics: **Effectiveness:** *The communication structure must effectively produce a qualified solution for the given task $q$.* **Cost-efficiency:** *The communication structure should minimize the overall resource consumption (e.g., model usage, API calls, token cost) required to solve $q$ under the given condition.* **Adaptiveness:** *The communication structure should dynamically adjust to varying conditions, ensuring robustness across diverse availability.*

---

We address this gap by formalizing the Adaptive Multi-Agent Communication Protocol (AMACP) and instantiating it via the *Conditional Agentic gRaph Designer* (CARD). CARD is a conditional graph-generation framework that (i) represents each agent via profile and condition channels, (ii) encodes dynamic environment signals, and (iii) decodes an interaction graph whose edges adapt at training time and at runtime as conditions change without retraining. The objective balances task utility and condition-aware communication cost, enforcing effectiveness, cost-efficiency, and adaptiveness required by AMACP. Empirically, on HUMANEVAL, MATH, and MMLU with simulated environmental changes (model upgrades, tool availability, data-source perturbations), CARD yields substantial gains over static and prompt-only or naively learned topologies while remaining competitive in static regimes. Our primary contributions are:

- Formalization of **AMACP**, a protocol enabling adaptive multi-agent communication under dynamic external conditions.

- Introduction of **CARD**, a conditional graph-generation framework explicitly learning effective and adaptive agent topologies from environmental states.

- Comprehensive empirical validation demonstrating that CARD consistently outperforms existing fixed and learned topology baselines under dynamic conditions.

- Detailed analyses of topology adaptations, elucidating how environmental state conditioning enhances the efficiency and robustness of multi-agent coordination.

## 2 RELATED WORK

**Collaborative LLM Agents.** Early work on LLM-based multi-agent communication has relied on manually defined coordination pipelines, with ranging from non-interactive queries and chain-of-thought prompting to debate frameworks and fixed tree- or graph-based structures (Wei et al.,

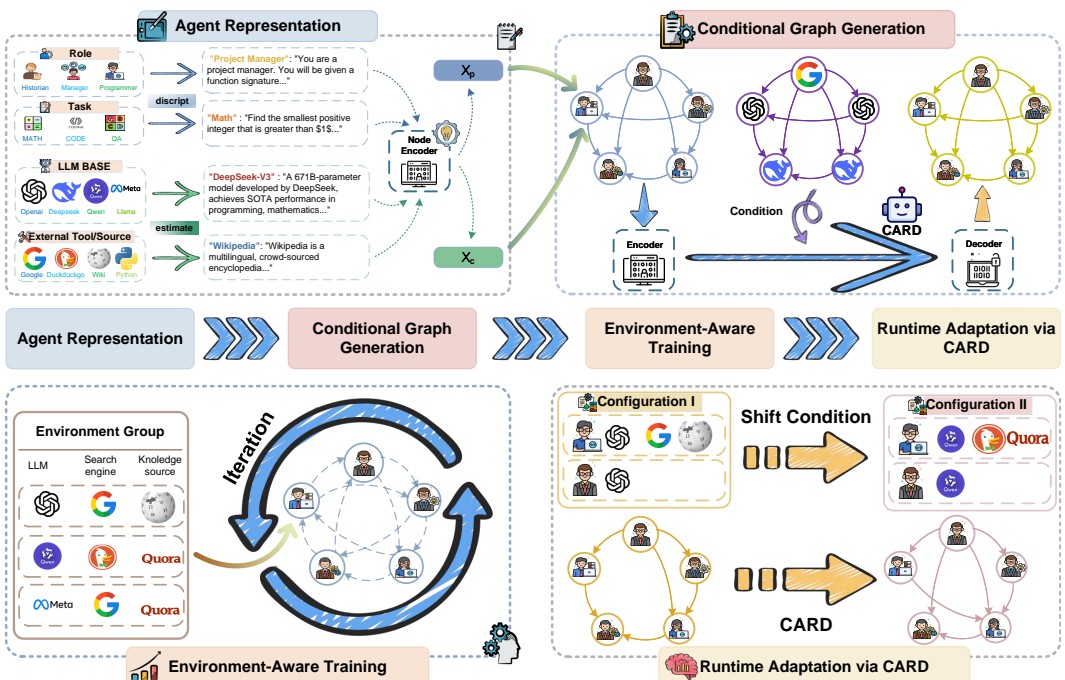

Figure 2: Overview of the Conditional Agentic Graph Designer (CARD) framework. Agent profiles and dynamic environment conditions are encoded into embeddings, which a conditional graph-generation module (Encoder → Condition Adaptation → Decoder) uses to produce an adaptive communication topology. CARD then performs environment-aware training, iteratively refining graphs under changing resource configurations, and deploys runtime adaptation to automatically update the multi-agent topology in response to new environmental states.

2022; Yao et al., 2023; Besta et al., 2024). To reduce the effort of handcrafting these pipelines, automated topology-learning methods such as GPT-Swarm (Zhuge et al., 2024), G-Designer (Zhang et al., 2025a), and Aflow (Zhang et al., 2025b) have been developed. These approaches optimize agent connections via differentiable modules or heuristic search, yielding strong performance in static settings. However, they continue to assume a stationary environment (Hu et al., 2025; Wang et al., 2025) and lack mechanisms to respond to changes in model capabilities, tool access, or data quality (Wu et al., 2024a).

**Multi-Agents as Graphs.** Although a few dynamic communication protocols have been proposed in distributed-systems literature (Wei et al., 2022; Yao et al., 2023; Liang et al., 2024), most learned topologies remain static and brittle under evolving conditions, such as model upgrades, fluctuations in external tool reliability, or shifts in data-source quality (Pareja et al., 2020; Chang et al., 2020). In these scenarios, pre-defined or naively optimized graphs can produce redundant interactions or disrupted information flows (Liu et al., 2025). To bridge this gap, we introduce the Conditional Agentic Graph Designer, which explicitly conditions graph generation on external signals (e.g., model version, tool performance, data-source fidelity) to produce adaptive, robust multi-agent topologies .

## 3 PROBLEM FORMULATION

We begin by formalizing the topology and protocol design space for LLM-based multi-agent systems (MAS), grounding the CARD framework (Figure 2) in well-defined constructs.

**Topological Structure of LLM-based Multi-Agent Systems.** A multi-agent system is represented as a directed graph $\mathcal{G} = (\mathcal{V}, \mathcal{E})$, where each node $v_i \in \mathcal{V}$ denotes an LLM-based agent and each directed edge $(v_i, v_j) \in \mathcal{E}$ represents a communication path from $v_i$ to $v_j$. Each agent $v_i$ is described by: profile attributes $P_i$, including [role identity, model base, tool access,

`historical state`]; And condition attributes $C_i$, capturing runtime environmental conditions. We model the condition $C$ as a composition of multiple features, where each feature corresponds to a distinct semantic aspect $\mathcal{F}$ such as model type, tool availability, or task complexity. Formally,

$$C = \{c_1, c_2, \ldots, c_k\}, \quad c_i \in \mathcal{F}_i, \tag{1}$$

$$\mathcal{C} = \mathcal{F}_1 \times \mathcal{F}_2 \times \cdots \times \mathcal{F}_k, \tag{2}$$

To ensure semantic consistency across heterogeneous features, we encode the structured condition using a unified pretrained language model, aligning all feature dimensions into a shared embedding space, enabling the model to handle unseen combinations of features during inference.

## 3.1 COMMUNICATION PIPELINE

Given a user query $\mathcal{Q}$, our system executes $K$ rounds of communication across a multi-agent topology $\mathcal{G} = (\mathcal{V}, \mathcal{E})$, where $\mathcal{V} = \{v_1, \ldots, v_N\}$ is the set of agents and $\mathcal{E}$ represents directed communication edges.

**Topological Scheduling.** To ensure valid information flow, a topological scheduling function $\varphi$ determines a permutation $\sigma = \{v_{(1)}, \ldots, v_{(N)}\}$ of agents that respects acyclic dependencies:

$$\varphi : \mathcal{G} \to \sigma, \quad \text{such that} \quad \forall j > i, \ v_{(j)} \notin \mathcal{N}_{\text{in}}(v_{(i)}), \tag{3}$$

where $\mathcal{N}_{\text{in}}(v_{(i)})$ denotes the set of upstream neighbors of agent $v_{(i)}$.

**Message Propagation.** At each communication round $t \in \{1, \ldots, K\}$, each agent $v_i$ receives (i) a system-level prompt $\mathcal{P}_{\text{sys}}^{(t)}$, (ii) a user-level prompt $\mathcal{P}_{\text{usr}}^{(t)}$, and (iii) the collection of responses from its incoming neighbors at the same round:

$$\mathcal{R}_i^{(t)} = v_i\left(\mathcal{P}_{\text{sys}}^{(t)}, \mathcal{P}_{\text{usr}}^{(t)}, \{\mathcal{R}_j^{(t)} : v_j \in \mathcal{N}_{\text{in}}(v_i)\}\right), \tag{4}$$

where $\mathcal{R}_i^{(t)}$ is the response generated by agent $v_i$ at round $t$.

**Output Aggregation.** After $K$ rounds of interaction, the system aggregates the final-round outputs from all agents to form the final system response:

$$\alpha^{(K)} = \text{Aggregate}\left(\mathcal{R}_1^{(K)}, \ldots, \mathcal{R}_N^{(K)}\right), \tag{5}$$

where $\text{Aggregate}(\cdot)$ denotes a task-specific aggregation function (e.g., voting, selection, or summarization) over the terminal responses.

## 3.2 AMACP: ADAPTIVE MULTI-AGENT COMMUNICATION PROTOCOL

To ensure meaningful topology construction, we define AMACP:

---
**AMACP Definition**

*Given a query $\mathcal{Q}$, a communication topology $\mathcal{G}$ must satisfy: 1. **Effectiveness:** Maximize conditioned task utility $u(\mathcal{G}(\mathcal{Q}|\mathcal{C})$; 2. **Cost-efficiency:** Minimize conditioned financial cost $w(\mathcal{G}; \mathcal{C})$; 3. **Adaptiveness:** Adjust topology in response to environmental condition $\mathcal{C}$ shifts. These objectives are jointly encoded in the following optimization problem:*

$$\min_{\mathcal{G} \in \mathbb{G}} \ \mathcal{L}_{\text{AMACP}}(\mathcal{G}; \mathcal{Q}, \mathcal{C}) = -u(\mathcal{G}(\mathcal{Q} \mid \mathcal{C})) + \beta \cdot w(\mathcal{G}; \mathcal{C}), \tag{6}$$

*where $u(\cdot)$ denotes the task-specific utility function, $w(\cdot)$ is the conditioned communication cost, and $\beta \in \mathbb{R}^+$ is a tunable trade-off hyperparameter.*

---

## 4 CONDITIONAL AGENTIC GRAPH DESIGNER

We introduce Conditional Agentic Graph Designer (CARD) that constructs adaptive, environment-conditioned multi-agent topologies. CARD comprises four key stages: (1) Agent representation, (2) Conditional graph generation, (3) Environment-aware training, and (4) Runtime adaptation. The complete workflow is summarized in Algorithm 1 and visualized in Figure 2.

**Agent Representation.** Given a query $Q$ and an environment configuration $C$, CARD first constructs an initial multi-agent network. Each agent $v_i \in V$ is described by two components: Firstly, a profile vector $P_i = [\mathcal{T}_p(\texttt{Base}_i), \texttt{Role}_i, \mathcal{T}_p(\texttt{Plugin}_i)]$, capturing static attributes of the agent, including its base model, assigned role, and supported tools. Here, $\mathcal{T}_p(\cdot)$ denotes a natural-language template function used to verbalize categorical features (e.g., model name, role identity, plugin type) into a text embedding. Secondly, a condition vector $C_i = \mathcal{T}_c(\mathcal{C}_i)$, describing runtime environment status for $v_i$, such as model availability, token cost, or API reliability. The function $\mathcal{T}_c(\cdot)$ generates textual descriptions that encode dynamic system conditions. These representations are later encoded as node features for conditional graph generation. See Appendix F for template instantiations of $\mathcal{T}_p$ and $\mathcal{T}_c$.

**Conditional Graph Generation** Given a user query $\mathcal{Q}$ and an initial environment configuration, CARD constructs a preliminary agent graph $\widetilde{\mathcal{G}} = (\mathcal{V}, \widetilde{\mathcal{E}})$ over $N = |\mathcal{V}|$ agents. Each agent $v_i \in \mathcal{V}$ is associated with a profile text $\mathcal{P}_i$ and a condition text $\mathcal{C}_i$, which are embedded as $X_i^p$ and $X_i^c$, respectively. Stacking across agents yields $X_p = [X_1^p, \ldots, X_N^p]$ and $X_c = [X_1^c, \ldots, X_N^c]$. The edge set $\widetilde{\mathcal{E}}$ is initialized from an anchor topology $\mathcal{A}$ (e.g., chain or star), which provides structural priors for initial connectivity.

To obtain a refined, query- and context-aware communication topology, CARD applies an encoder–decoder graph generation module. The encoder comprises two learnable graph encoders, $\phi_p$ and $\phi_c$, that produce latent representations for profile and condition channels:

$$\mathbf{H}_p = \phi_p(H_p \mid X_p, \mathcal{A}; \Theta_p), \tag{7}$$

$$\mathbf{H}_c = \phi_c(H_c \mid X_c, \mathcal{A}; \Theta_c), \tag{8}$$

where $\mathbf{H}_p = [\mathbf{h}_1^p, \ldots, \mathbf{h}_N^p]$ and $\mathbf{H}_c = [\mathbf{h}_1^c, \ldots, \mathbf{h}_N^c]$ denote the latent states, and $\Theta_p, \Theta_c$ are encoder parameters. The decoder $\psi_\theta$ then estimates pairwise edge probabilities conditioned on these latent states and a query embedding $\mathbf{h}_\mathcal{Q}$ (the query is treated as an auxiliary node that attends to all agents in both channels):

$$\psi(S \mid \mathbf{H}_p, \mathbf{H}_c) = \prod_{i,j} \psi\left(S_{ij} \mid \mathbf{h}_i^p, \mathbf{h}_i^c, \mathbf{h}_j^p, \mathbf{h}_j^c, \mathbf{h}_\mathcal{Q}; \Theta_d\right), \tag{9}$$

where $S_{ij} \in [0, 1]$ is the predicted link probability and $\Theta_d$ are decoder parameters. Finally, the communication topology is obtained by thresholding the predicted adjacency:

$$\mathcal{E}_{\text{com}} = \{(v_i, v_j) \mid S_{ij} > \tau\}, \qquad \mathcal{G}_{\text{com}} = (\mathcal{V}, \mathcal{E}_{\text{com}}), \tag{10}$$

with a user-specified or validation-selected threshold $\tau \in (0, 1)$. The resulting $\mathcal{G}_{\text{com}}$ serves as the backbone for downstream multi-agent communication and reasoning, adaptively modulated by static profiles and dynamic runtime states.

**Environment-Aware Training.** Given a query $Q$ and an environment condition $\mathcal{C}$ (Section 3), CARD trains by iterating over sampled $(Q, \mathcal{C})$ pairs and running $K \in \mathbb{N}$ rounds of multi-agent interaction on $\mathcal{G}_{\text{com}}$. At communication round $t \in \{1, \ldots, K\}$, agent $v_i$ receives a system-level prompt $\mathcal{I}_{\text{sys}}^{(t)}$, a user-level prompt $\mathcal{I}_{\text{usr}}^{(t)}$, and upstream messages $\{\mathcal{R}_j^{(t)} \mid v_j \in \mathcal{N}_{\text{in}}(v_i)\}$, and produces a response $\mathcal{R}_i^{(t)}$ with equation 4. And after $K$ rounds, a task-specific aggregation operator $\text{AGGREGATE}(\cdot)$ (e.g., voting, selection, or summarization) combines terminal responses into the system output $\alpha^{(K)}$ with equation 5.

Let $\Theta_p, \Theta_c, \Theta_d$ denote the parameters of the profile encoder $\phi_p$, condition encoder $\phi_c$, and graph decoder $\psi$. We optimize these parameters by gradient descent on a CARD loss that instantiates the AMACP objective (Eq. equation 6):

$$\mathcal{L}_{\text{CARD}}(Q, \mathcal{C}; \Theta_p, \Theta_c, \Theta_d) = -\underbrace{u\left(\alpha^{(K)}\right)}_{\text{task utility}} + \beta \underbrace{w(\mathcal{G}_{\text{com}}; \mathcal{C})}_{\text{condition-aware cost}}, \tag{11}$$

where $u(\cdot)$ is a task-specific utility (e.g., accuracy/probability of correctness), $w(\cdot)$ measures the conditioned communication cost, and $\beta > 0$ balances utility and cost. Specifically, to encourage communication efficiency while preserving performance, we regularize the *soft* communication

graph $\widetilde{\mathcal{G}}_{\text{com}}$ output by the decoder. Let $S \in [0,1]^{N \times N}$ be the predicted (directed) link-probability matrix and define $p_{ij} := S_{ij}$ as the probability that edge $(v_i \to v_j)$ is active under condition $\mathcal{C}$. Let $\text{Cost}_{ij} \geq 0$ denote the expected token-level inference cost on edge $(i,j)$ (a function of the base model(s) and the number of exchanged tokens). The condition-aware regularizer is:

$$\min_{\widetilde{\mathcal{G}}_{\text{com}} \in \mathbb{G}} w\left(\widetilde{\mathcal{G}}_{\text{com}}, \mathcal{C}\right) = \sum_{(i,j) \in \widetilde{\mathcal{G}}_{\text{com}}} \text{Cost}_{ij}\, p_{ij}, \tag{12}$$

where $\mathbb{G}$ is the space of admissible (soft) directed graphs over $\mathcal{V}$. In practice, $\mathcal{G}_{\text{com}}$ used for execution is obtained by thresholding $S$; training backpropagates through $S$ to update $(\Theta_p, \Theta_c, \Theta_d)$ via equation 11.

**Runtime Adaptation via CARD.**    At deployment, when external conditions change (e.g., base model capability, tool reliability, or cost), CARD updates the communication topology without retraining by decoding new edges from refreshed condition signals:

$$\mathcal{G}_{\text{com}}^{\text{new}} = \psi\left(\phi_p(X_p),\, \phi_c(X_c^{\text{new}}),\, \mathcal{A}\right), \tag{13}$$

where $X_p$ encodes the agent profiles *static* (role, base model, and tools), $X_c^{\text{new}}$ encodes the runtime conditions *updated*, $\phi_p, \phi_c$ maps these to latent node states, $\mathcal{A}$ is the anchor prior (e.g. chain, star, or fully connected), and $\psi$ decodes edge probabilities (thresholded at $\tau$) to produce the revised adjacency. This one-pass recomputation preserves robust, cost-efficient collaboration under real-time shifts.

## 5    EXPERIMENT

**Datasets and Metrics.**    We assess CARD on three standard benchmarks: programming code generation (HumanEval)(Chen et al., 2021), mathematical reasoning (MATH)(Hendrycks et al., 2021b), and general reasoning and language understanding (MMLU)(Hendrycks et al., 2021a).

**Baselines and Setup.**    We compare our approach against three categories of methods: **Vanilla LLM**, using the model's native capabilities to produce direct answers; **Manually designed agents**, including Chain-of-Thought (CoT)(Wei et al., 2022) in a single-agent setup, and *LLM-Debate*(Du et al., 2024) and *Random Graph* in a multi-agent configuration; And **Automatically optimized topologies**, graph-learning techniques such as *GPT-Swarm*(Zhuge et al., 2024) and *G-Designer*(Zhang et al., 2025a) (which share our graph formulation), alongside the heuristic rule-based optimizer *Aflow*(Zhang et al., 2025b). We evaluate a diverse set of language models sourced from different providers, each representing distinct technical paradigms, training methodologies, and architectural designs. Please see Appendix C and  G for more details.

### 5.1    MAIN RESULTS

**Conditional design (CARD) consistently delivers the best overall performance.**    With 90.50% on HumanEval, 74.50% on MATH, and 86.67% on MMLU, remarkably, CARD attains or ties for the top score in 13 out of 15 model–benchmark combinations, demonstrating strong robustness across different LLM bases.

**Gains accrue progressively with richer design abstractions.**    Single-agent methods (Vanilla, CoT) establish a competitive baseline but lack collaboration. Fixed multi-agent topologies (Random-graph, LLM-Debate) add modest gains (+0.5–2.0 pp) by enabling parallel reasoning. Automated topology learners (GPT-swarm, Aflow, G-designer) further boost performance (+1.0–4.0 pp) by optimizing static communication structures. Finally, CARD's conditional adaptation delivers an extra +0.5–3.0 pp advantage over these static designs by tailoring the topology to environmental signals.

**Conditional adaptation pays off especially under out-of-domain settings.**    By explicitly conditioning on model- and tool-state, CARD narrows the gap between in-domain and out-domain evaluations. For example, on MATH, G-designer's accuracy falls from 91.66% to 79.16% when changing from deepseek-v3 to qwen-72B, whereas CARD's drop is smaller (from 91.66% to 82.50%), underscoring its adaptability to unseen settings.

Table 1: Evaluation of multi-agent and topology design methods on HumanEval, MATH, and MMLU. "Mul.", "Auto.", and "Cond." indicate support for multi-agent collaboration, automated topology design, and conditional configuration, where ✓ indicates "Yes" and ✗ indicates "No". For automated methods, CYAN cells denote in-domain adaptation (trained and tested on the same LLM), GREEN cells denote out-domain adaptation (generalization to unseen LLMs), and YELLOW cells denote the average performance.

| Method | Mul. | Auto. | Cond. | gpt4o-mini | deepseek-v3 | llama3-70B | gpt4o | qwen-72B | Avg. |
|---|---|---|---|---|---|---|---|---|---|
| | | | | | HumanEval | | | | |
| Vanilla | ✗ | ✗ | ✗ | 85.83 | 92.50 | 76.66 | 85.83 | 86.66 | 85.50 |
| CoT | ✗ | ✗ | ✗ | 88.33 | 93.33 | 78.33 | 90.00 | 88.33 | 87.66 |
| Random-graph | ✓ | ✗ | ✗ | 87.50 | 94.16 | 73.33 | 90.00 | 85.00 | 86.00 |
| LLM-Debate | ✓ | ✗ | ✗ | 91.66 | 95.00 | 75.83 | 87.50 | 80.00 | 86.00 |
| GPTswarm | ✓ | ✓ | ✗ | 89.16 | 92.50 | 78.33 | 91.66 | 81.66 | 86.66 |
| Aflow | ✓ | ✓ | ✗ | 90.83 | 92.50 | **85.83** | 93.33 | 86.66 | 89.83 |
| G-designer | ✓ | ✓ | ✗ | 89.16 | 94.16 | 75.00 | 88.33 | 85.83 | 86.50 |
| **CARD** | ✓ | ✓ | ✓ | **93.33** | **95.83** | 81.66 | **93.33** | **88.33** | **90.50** |
| | | | | | MATH | | | | |
| Vanilla | ✗ | ✗ | ✗ | 59.16 | 74.16 | 41.66 | 70.00 | 71.66 | 63.33 |
| CoT | ✗ | ✗ | ✗ | 61.66 | 80.00 | 41.66 | 65.00 | 73.33 | 64.33 |
| Random-graph | ✓ | ✗ | ✗ | 60.00 | 88.33 | 38.33 | 70.00 | 66.67 | 64.67 |
| LLM-Debate | ✓ | ✗ | ✗ | 59.16 | 85.00 | 46.66 | 71.66 | 71.66 | 66.83 |
| GPTswarm | ✓ | ✓ | ✗ | 67.50 | 90.83 | 46.66 | 67.50 | 78.33 | 70.16 |
| Aflow | ✓ | ✓ | ✗ | **80.83** | 91.66 | 40.00 | 75.83 | 80.83 | 73.83 |
| G-designer | ✓ | ✓ | ✗ | 70.00 | 91.66 | 47.50 | 75.00 | 79.16 | 72.66 |
| **CARD** | ✓ | ✓ | ✓ | 73.33 | **91.66** | **48.33** | **76.67** | **82.50** | **74.50** |
| | | | | | MMLU | | | | |
| Vanilla | ✗ | ✗ | ✗ | 77.12 | 86.27 | 75.16 | 86.93 | 79.74 | 81.04 |
| CoT | ✗ | ✗ | ✗ | 81.05 | 92.16 | 75.16 | 88.89 | 83.66 | 84.18 |
| Random-graph | ✓ | ✗ | ✗ | 79.74 | 90.85 | 75.82 | 86.93 | 83.00 | 83.27 |
| LLM-Debate | ✓ | ✗ | ✗ | 80.39 | 90.20 | 76.47 | 88.24 | 84.31 | 83.92 |
| GPTswarm | ✓ | ✓ | ✗ | 82.35 | 89.54 | 77.78 | 86.27 | 84.31 | 84.05 |
| Aflow | ✓ | ✓ | ✗ | 79.74 | 84.97 | 77.12 | 89.54 | 83.00 | 82.87 |
| G-designer | ✓ | ✓ | ✗ | 83.00 | 90.85 | 77.12 | 86.93 | 84.31 | 84.44 |
| **CARD** | ✓ | ✓ | ✓ | **84.97** | **93.46** | **80.39** | **89.54** | **84.97** | **86.67** |

## 5.2 How to Embed Conditions in LLM Topology Generation?

Figure 3 reports an ablation study on different ways to inject environmental conditions into the generation of multi-agent topologies. We compare an unconditioned baseline (w/o Cond.), a naive prompt-level injection (w/ Cond.p) that appends condition descriptors to the system prompt, and our CARD approach which embeds conditions directly within the graph-generation module. Each variant is evaluated in HumanEval, MATH, and MMLU on five LLM bases, reporting absolute precision and $\Delta$ precision relative to the unconditioned baseline.

**CARD delivers robust, non-negative gains across all benchmarks.** While simple prompt conditioning can backfire, causing up to a –12.50 % drop on MATH with base M5 and –2.00 % on MMLU with base M1. CARD consistently yields positive improvements on every model–benchmark pair (e.g., +0.83 % to +3.34 % on MATH, +0.66 % to +2.62 % on MMLU, and +2.50 % to +23.33 % on HumanEval), proving structured topology adaptation far more reliable than prompt-only methods.

**CARD compensates for weaker baseline models.** The greatest uplifts appear in the most challenging settings, such as a +2.62 % gain on MMLU with M3 and a +3.34 % boost on MATH with M5, demonstrating the ability of CARD to narrow out-of-domain performance gaps and mitigate the limitations of less capable LLM bases. Further analyses isolating source/tool effects and localized condition perturbations are in Appendix B.2.

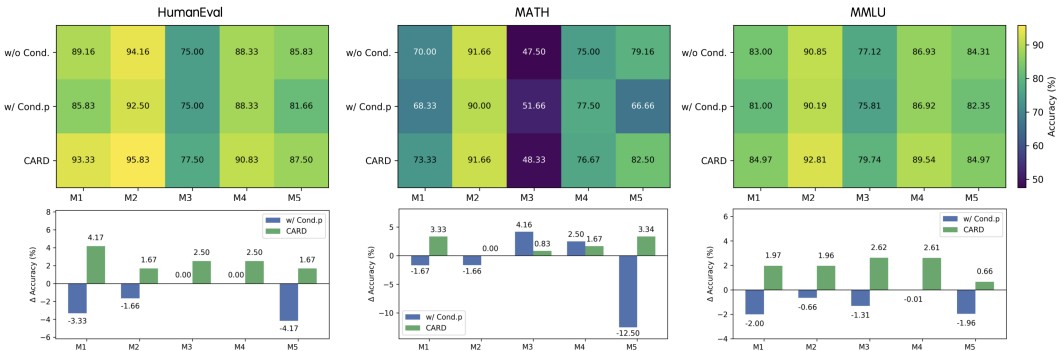

Figure 3: Performance and gains of w/o Cond., w/ Cond.p, and CARD on HumanEval, MATH, and MMLU across LLM bases (M1–M5, same to Table 1). **Top:** absolute accuracy (%). **Bottom:** $\Delta$ accuracy (%) over the w/o Cond. baseline.

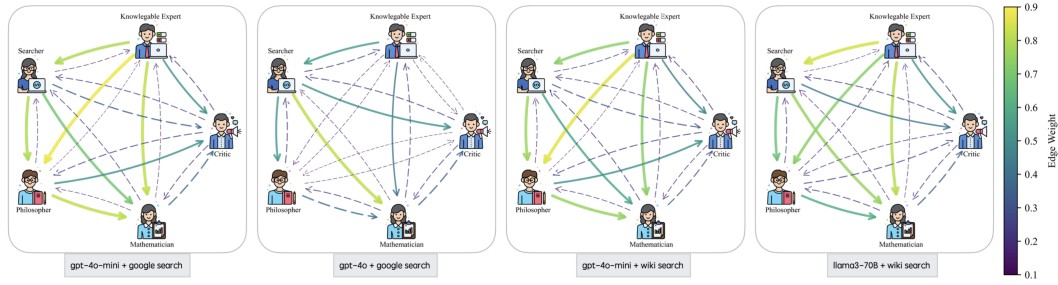

Figure 4: Visualization of CARD topology matrices (See Appendix D for matrices and correlation analysis details.) under different conditions. Edge thickness reflects the communication probability between agents. Configurations 1 to 4 (Table 10) are shown from left to right.

### 5.3 How Do Environmental Conditions Shape the Final Communication Architecture?

In Figure 4, we present four representative experimental configurations, combining two LLM capacities (GPT-4o-mini vs. GPT-4o or Llama3-70B) with two search engines (Google Search vs. Wiki Search) to illustrate CARD's conditional adaptation in action. By visualizing the resulting topology matrices, we aim to quantify how variations in model strength and retrieval quality drive changes in edge density, directional flow patterns, and overall graph structure. This case study validates CARD's ability to tailor multi-agent communication graphs to dynamic environmental signals, providing insight into the practical behavior of the protocol under realistic operational changes.

**Weaker Models Demand Denser Collaboration.** In Configuration 1 (GPT-4o-mini + Google Search), the average edge weight is substantially greater than in Configuration 2 (GPT-4o + Google Search), demonstrating that the smaller model compensates for lower inherent capacity by intensifying multi-agent communication.

**Search-Engine Swap Preserves Global Structure but Shifts Local Flows.** Replacing Google with Wiki for GPT-4o-mini (Config 1 vs. 3) yields a Pearson correlation of $r = 0.9797$ and $p = 0.0006$ (Appendix D.2), confirming near-identical overall topology. Locally, however, the *Knowledge Expert→Searcher* edge weight decreases markedly, and the *Searcher→Mathematician* link drops more than *Searcher→Philosopher*, reflecting domain-specific retrieval efficacy differences.

**Lowest Capacity + Lower-Quality Search Maximizes External-Knowledge Reliance.** The Llama3-70B + Wiki configuration (Config 4) produces the densest graph with the highest average edge weights, demonstrating peak dependence on external information when both model capacity and search quality are reduced (Config 1 vs. 4: $r = 0.7789$, $p = 0.0679$ (Appendix D.2)).

We provide additional quantitative breakdowns by model capability and size in Appendix B.1 and by tools/knowledge sources in Appendix B.2. Scalability under varying agent counts is detailed in

Appendix B.3, and robustness under targeted attacks and accuracy–cost trade-offs are summarized in Appendix B.4.

## 6 CONCLUSION

We introduced AMACP, a protocol for adaptive multi-agent communication, and CARD, a conditional graph-generation framework that tailors LLM-based agent topologies to dynamic environments. Experiments on HumanEval, MATH, and MMLU under simulated shifts (model upgrades, tool changes, and data perturbations) show CARD outperforms static and prompt-based designs by up to three percentage points in accuracy while remaining cost-effective. Topology visualizations underscore CARD's capability to adjust communication patterns based on agent capabilities and resource quality. Future work will scale to larger agent ensembles, integrate online reinforcement for continual adaptation, and validate CARD in real-world multi-agent applications. For a detailed discussion of limitations and avenues for future work, please refer to Appendix A.

## 7 ETHICS STATEMENT

This work uses only publicly available datasets and models, and does not involve human subjects or private data. We acknowledge the broader societal risks of autonomous multi-agent LLM systems and encourage responsible deployment with appropriate safeguards.

## 8 REPRODUCIBILITY STATEMENT

To support reproducibility, we provide the full source code, training and evaluation scripts, and prompt templates at `https://anonymous.4open.science/r/agentgraph-FF9A`. All experiments are based on publicly available benchmarks (HumanEval, MATH, MMLU) and open-source or API-accessible LLMs, with full implementation details, model configurations, and hyperparameters documented in Appendices C–F. Results are averaged over multiple runs, and all metrics and visualizations are script-generated for easy verification.

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

APPENDIX CONTENTS

## A    LIMITATION

This work focuses on conditional optimization and adaptation of multi-agent communication topologies. However, it does not explicitly update agent-level configurations such as individual prompts or internal profiles in response to environmental shifts. In practice, jointly optimizing both the communication topology and agent behaviors, including prompt augmentation and tool selection strategies, may further improve system performance. Exploring this direction remains an open avenue for future research. Additionally, while our formulation represents the multi-agent system as a graph, which offers a cognitively interpretable and analyzable abstraction, graph-based representations may be insufficient for capturing domain-specific nuances in complex real-world settings. For example, in software engineering workflows, procedural constraints, tool dependencies, and execution semantics are often critical. Future work may incorporate human-in-the-loop expertise, such as software development best practices and debugging heuristics, and explore hybrid models that combine symbolic priors with learned agent adaptation mechanisms.

## B    ANALYSIS AND DISCUSSION

### B.1    QUANTITATIVE ANALYSIS OF CONDITIONS: MODEL SIZE AND REASONING ABILITY

We conduct further experimental analysis to investigate how variations in model capability, model size, external tools, and knowledge sources impact multi-agent topology design and overall performance.

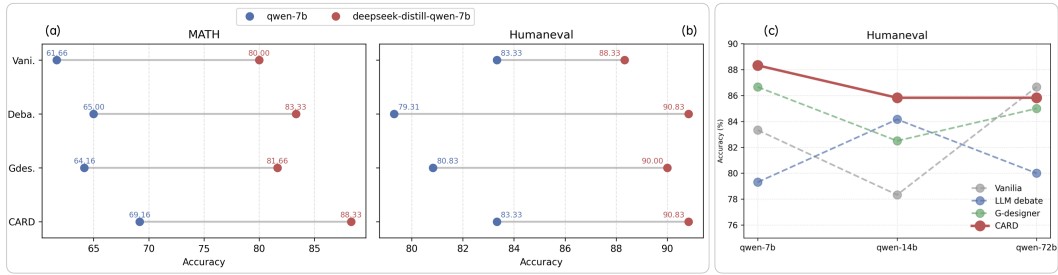

Figure 5: **Left**: Accuracy on MATH and HumanEval across LLMs with different reasoning capabilities. **Right**: Accuracy across different model sizes within the same LLM family.

**Stronger and larger base models yield higher multi-agent performance, with CARD amplifying these gains, but on simple tasks a single powerful LLM can outperform multi-agent coordination due to communication overhead**    On both MATH and HumanEval benchmarks, upgrading from qwen-7b to the higher-capability deepseek-distill-qwen-7b yields consistent accuracy improvements across all methods, with CARD showing the largest absolute gains and the best performance(e.g., MATH accuracy rises from 69.16 % to 88.33 % (+19.17 pp)). However, on simple benchmarks with a capable model (Figure 5 Right), vanilla single-agent slightly outperforms multi-agent due to redundant communication. This underscores that multi-agent benefits require task complexity to outweigh coordination costs.

**CARD exhibits superior robustness to variations in external tools and knowledge resources** Evidence: When switching among Google Search, DuckDuckGo, and Wikipedia as the external tool, CARD's HumanEval accuracy only drops from 85.62 % to 83.00 % Figure 6(Left), a smaller decline than alternative methods. Similarly, across knowledge sources (Wikipedia, Tutorialspoint, Quora), CARD achieves its highest performance (85.62 %) on the richest data source and outperforms other approaches by 1–2 percentage points even on less informative sources. This resilience highlights CARD's ability to maintain strong multi-agent topologies under diverse resource conditions.

## B.2  Quantitative Analysis of Conditions: Tools and Knowledge Resources

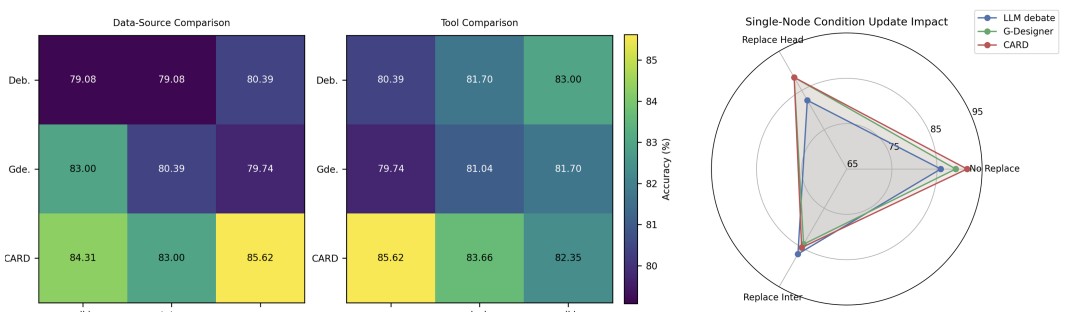

Figure 6: **Left**: Performance comparison using different external knowledge sources. **Central**: Performance across various available tools (search engines). **Right**: Impact on multi-agent performance on HumanEval, when only a single node's condition is updated instead of the global agent condition.

We evaluate how external conditions, namely knowledge sources and retrieval tools, affect performance in the CARD framework. Experiments use HumanEval with the same base LLM and agent-role design as in the main paper. The only varying factors are the condition features: switching among three knowledge corpora (Wikipedia, Tutorialspoint, Quora), changing search tools (Google, DuckDuckGo, WikiSearch), and applying localized perturbations by modifying a single node's condition embedding. All configurations share the same retrieval budget, prompt formatting, and random seed.

Across source and tool variations, CARD consistently outperforms baselines and exhibits smaller performance drops under weaker conditions. Accuracy declines from 85.6% on Wikipedia to 83.0% on Quora, yet CARD maintains a clear margin over LLM-Debate and G-Designer. This resilience stems from condition-aware graph generation that adapts edge density and agent coordination to upstream content quality. The results indicate that topology-level adaptation provides stronger robustness than prompt-only conditioning when facing domain shift or noisy external knowledge.

In local perturbation tests, changing the condition for only one node, either the root or an intermediate node, preserves most performance at 88.3% and 85.0% respectively, substantially surpassing baselines. This shows that CARD enables low-cost, localized reconfiguration without full graph retraining. The modular design supports practical online adaptation in production, allowing lightweight updates in response to tool variability or hotfixes while maintaining stable accuracy under dynamic real-world constraints.

### B.3  Multi-agent Scalability Analysis

We evaluate the scalability and robustness of CARD against G-Designer by grouping base LLMs into in-domain (`gpt4o-mini`, `deepseek-v3`, `llama3`) and out-of-domain (`GPT-4o`, `qwen-72B`) settings.

**CARD scales more effectively than G-Designer as agent count increases, especially in out-of-domain settings.** As shown in Figure 7, CARD consistently achieves higher MMLU scores as the number of agents increases, with particularly pronounced gains in the out-of-domain setting (up to +1.99 pp over G-Designer at 10 agents). In the in-domain case, both methods improve over Vanilla, but CARD shows a steeper upward trend, with its advantage widening at 5–10 agents. This indicates that conditional topology generation in CARD helps agents coordinate more effectively as system size grows.

**CARD also demonstrates stronger robustness under domain shift with comparable uncertainty.** The method also exhibits greater robustness under distribution shift. While G-Designer's performance gains plateau in the out-of-domain setting, CARD continues to benefit from agent scaling. Moreover, CARD achieves these improvements with comparable or slightly lower confidence interval widths, suggesting more reliable and generalizable coordination gains. These results align with the design goal of CARD—namely, to generate topology conditioned on external constraints (e.g., model

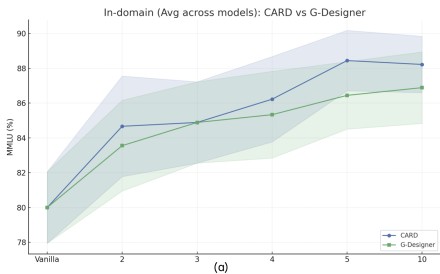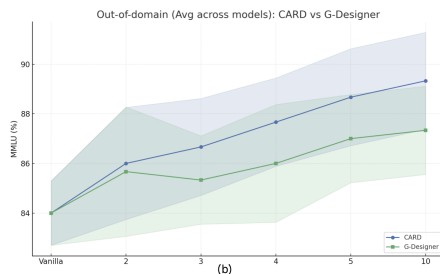

Figure 7: MMLU performance of CARD vs G-Designer across varying agent counts. CARD consistently outperforms G-Designer in both (a) in-domain and (b) out-of-domain settings, with larger gains under domain shift; shaded areas denote 95% confidence intervals.

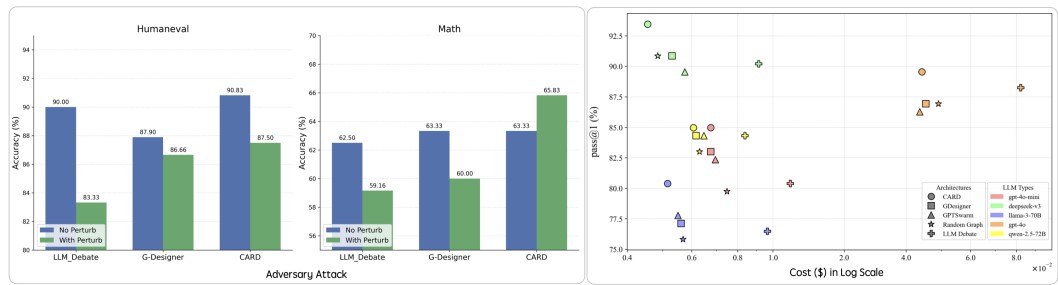

Figure 8: **Left**: Accuracy comparison before and after adversarial attacks across different methods. **Right**: The inference cost (USD per instance) across methods and LLMs.

type, cost, capability), enabling adaptive and scalable multi-agent collaboration even in unseen environments.

## B.4 MULTI-AGENT ROBUSTNESS & COST-EFFICIENCY ANALYSIS

We evaluate both robustness and cost-efficiency by simulating targeted attacks and configuration faults at intermediate agents on the HumanEval benchmark to measure resilience in accuracy, and by calculating total training and evaluation expenses across different LLM bases to quantify economic trade-offs, thereby enabling a systematic comparison of static, learned, and conditionally adapted communication topologies under both adverse conditions and budget constraints.

**Robustness under attack is markedly improved by conditional adaptation.** As shown in Figure 8 (Left), when an agent node is attacked, LLM Debate, which relies on fixed pairwise prompting without structural adaptation, and suffers the sharpest performance drop (–6.67 pp on HumanEval, -3.34 pp on MATH). G-Designer, trained under attack conditions, filters out the faulty node and shows smaller degradation (–1.24 pp on HumanEval, –3.33 pp on MATH), but loses generalization once the node recovers. In contrast, CARD, trained under both attacked and clean conditions, not only outperforms G-Designer under attack (87.50% vs. 86.66% on HumanEval; 65.83% vs. 60.00% on MATH), but also shows significantly greater recovery when the compromised node is restored (+3.33 pp vs. +1.24 pp on HumanEval; the accuracy under attack is even higher than non-attacked on MATH, indicating that our topology is fully adaptable to both attacked and non-attacked conditions.) highlighting its superior resilience and adaptability across both degraded and recovered environments.

**Localized condition updates recover most of the lost performance at minimal adaptation cost.** Figure 6 (Right) illustrates that, upon replacing only a single node's condition rather than regenerating the entire communication graph, CARD retains 88.33 % accuracy under head-node perturbations and 85.00 % under intermediate-node perturbations, demonstrating overall superiority over both LLM-Debate and G-Designer, highlighting that fine-grained adaptation can preserve robustness with far lower computational overhead than global reconfiguration.

**Conditional designs deliver the best accuracy-to-cost balance among all methods.** In Figure 8 (Right), CARD's configurations (e.g., achieving 94 % accuracy at an evaluation cost of $4 \times 10^{-3}$ USD) occupy the upper–left region of the cost–performance plane, while static multi-agent schemes like LLM-Debate and learned topologies such as G-Designer generally cost more to reach a lower accuracy; this confirms that conditionally adapted graphs not only boost resilience but also minimize economic expenditure for a given performance level.

## C  IMPLEMENTATION DETAILS

Specifically, we include OpenAI's `GPT-4o`OpenAI (2024a) and `GPT-4o-mini`OpenAI (2024b), DeepSeek's `DeepSeek-V3`DeepSeek-AI et al. (2024) (630B), Meta's `Llama3-70B`Dubey et al. (2024), and the Qwen series from Alibaba, comprising `Qwen-2.5` models (7B, 14B, and 72B)Yang et al. (2024) as well as the distilled variant `qwen-distill-r1-7b`DeepSeek-AI et al. (2025) derived from `Qwen-2.5-7B`. These models differ across providers (capturing variations in technical trajectories and architectural choices), model sizes (correlating with computational capacity and performance), and domain specializations (affecting knowledge scope and inference behaviors). In addition, we examine the influence of external tools on multi-agent architectures by varying the underlying data sources and search engines. The search engines considered are `Google Search`, `DuckDuckGo`, and `Wiki Search`; the data sources include `Quora`, `Wikipedia`, and `Tutorialspoint`. These configurations represent diverse retrieval qualities and knowledge coverage levels. This setup is intended to emulate the dynamic evolution of resources in real-world multi-agent systems, enhancing the robustness and adaptability of learned multi-agent topologies under changing conditions.

## D  SOURCE DATA

### D.1  ADJACENCY MATRIX OF VISUALIZATION

Table 2: Adjacency Matrix 1: `gpt-4o-mini + Google`

|  | Knowlegable Expert | Searcher | Philosopher | Mathematician | Critic |
|---|---|---|---|---|---|
| **Knowlegable Expert** | Masked | 0.79 | 0.88 | 0.81 | 0.5 |
| **Searcher** | 0.17 | Masked | 0.79 | 0.69 | 0.34 |
| **Philosopher** | 0.11 | 0.17 | Masked | 0.81 | 0.5 |
| **Mathematician** | 0.15 | 0.21 | 0.15 | Masked | 0.37 |
| **Critic** | 0.25 | 0.22 | 0.25 | 0.23 | Masked |

Table 3: Adjacency Matrix 2: `gpt-4o + Google`

|  | Knowlegable Expert | Searcher | Philosopher | Mathematician | Critic |
|---|---|---|---|---|---|
| **Knowlegable Expert** | Masked | 0.54 | 0.15 | 0.4 | 0.14 |
| **Searcher** | 0.25 | Masked | 0.53 | 0.81 | 0.5 |
| **Philosopher** | 0.13 | 0.25 | Masked | 0.39 | 0.13 |
| **Mathematician** | 0.24 | 0.15 | 0.24 | Masked | 0.36 |
| **Critic** | 0.12 | 0.25 | 0.11 | 0.23 | Masked |

Table 4: Adjacency Matrix 3: `gpt-4o-mini + Wiki`

|  | Knowlegable Expert | Searcher | Philosopher | Mathematician | Critic |
|---|---|---|---|---|---|
| **Knowlegable Expert** | Masked | 0.73 | 0.86 | 0.73 | 0.46 |
| **Searcher** | 0.2 | Masked | 0.76 | 0.57 | 0.3 |
| **Philosopher** | 0.12 | 0.18 | Masked | 0.75 | 0.5 |
| **Mathematician** | 0.2 | 0.25 | 0.19 | Masked | 0.29 |
| **Critic** | 0.25 | 0.21 | 0.25 | 0.21 | Masked |

Table 5: Adjacency Matrix 4: `Llama-3-70B + Wiki`

|  | Knowlegable Expert | Searcher | Philosopher | Mathematician | Critic |
|---|---|---|---|---|---|
| **Knowlegable Expert** | Masked | 0.84 | 0.73 | 0.8 | 0.5 |
| **Searcher** | 0.13 | Masked | 0.67 | 0.75 | 0.42 |
| **Philosopher** | 0.2 | 0.22 | Masked | 0.6 | 0.27 |
| **Mathematician** | 0.16 | 0.19 | 0.24 | Masked | 0.35 |
| **Critic** | 0.25 | 0.24 | 0.2 | 0.23 | Masked |

## D.2 CORRELATION ANALYSIS OF ADJACENCY MATRIX

CORRELATION ANALYSIS

Table 6: Pearson correlation between matrix pairs with corresponding strength and significance.

| Comparison | $r$ | $p$ | Strength | Sig. |
|---|---|---|---|---|
| Matrix 1 vs 2 | 0.32 | 0.54 | Weak | No |
| Matrix 1 vs 3 | **0.98** | **0.001** | Very strong | Yes |
| Matrix 1 vs 4 | 0.78 | 0.07 | Strong | Marginal |

## E ALGORITHM WORKFLOW

---
**Algorithm 1** Workflow of **CARD**: Conditional Agentic Graph Designer
---
**Input:** Query set $\{\mathcal{Q}_1, \ldots, \mathcal{Q}_D\}$, condition configurations $\{\mathcal{C}_1, \ldots, \mathcal{C}_C\}$,
Graph auto-encoder $f_\nu = (q_{\text{stat}}, q_{\text{dyn}}, \psi)$ with parameters $(\Theta_p, \Theta_c, \Theta_d)$, learning rate $\alpha$

1: **for** each query $\mathcal{Q}_d \in \{\mathcal{Q}_1, \ldots, \mathcal{Q}_D\}$ **do**
2:     **for** each condition $\mathcal{C}_c \in \{\mathcal{C}_1, \ldots, \mathcal{C}_C\}$ **do**
3:         /* Construct agent features under condition $\mathcal{C}_c$ */
4:         **for** agent $v_i \in \{v_1, \ldots, v_N\}$ **do**
5:             $\mathbf{x}_i^p \leftarrow \mathcal{T}_p(\texttt{Base}_i, \texttt{Role}_i, \texttt{Plugin}_i)$
6:             $\mathbf{x}_i^c \leftarrow \mathcal{T}_c(\mathcal{C}_c[i])$
7:         **end for**
8:         $\mathbf{X}_p \leftarrow [\mathbf{x}_1^p, \ldots, \mathbf{x}_N^p]^\top, \quad \mathbf{X}_c \leftarrow [\mathbf{x}_1^c, \ldots, \mathbf{x}_N^c]^\top$
9:         $\mathbf{x}_{\mathcal{Q}} \leftarrow \text{Embed}(\mathcal{Q}_d)$           ▷ query treated as a virtual agent node
10:         Define anchor topology $\mathcal{A}$ (e.g., fully-connected + task node)
11:         $\widetilde{\mathcal{G}} \leftarrow (\{\mathbf{X}_p, \mathbf{X}_c, \mathbf{x}_{\mathcal{Q}}\}, \mathcal{A})$
12:         /* Generate communication topology via encoder-decoder */
13:         $\mathbf{H}_p \leftarrow \phi_p(\mathbf{X}_p \mid \mathcal{A}), \quad \mathbf{H}_c \leftarrow \phi_c(\mathbf{X}_c \mid \mathcal{A})$
14:         $\mathbf{S} \leftarrow \psi(\mathbf{H}_p, \mathbf{H}_c, \mathbf{x}_{\mathcal{Q}})$         ▷ compute link probabilities
15:         $\mathcal{G}_{\text{com}} \leftarrow \{(i, j) \mid S_{ij} > \tau\}$         ▷ retain edges above threshold
16:         /* Multi-agent collaboration under $\mathcal{G}_{\text{com}}$ */
17:         **for** $t = 1$ to $K$ **do**
18:             **for** agent $v_i$ in schedule $\phi(\mathcal{G}_{\text{com}})$ **do**
19:                 $\mathcal{P}_{\text{usr}}^{(t)} \leftarrow \{\mathcal{Q}_d\} \cup \{\mathcal{R}_j^{(t)} \mid v_j \in \mathcal{N}_{\text{in}}(v_i)\}$
20:                 $\mathcal{R}_i^{(t)} \leftarrow v_i(\mathcal{P}_{\text{sys}}^{(t)}, \mathcal{P}_{\text{usr}}^{(t)})$
21:             **end for**
22:             $\alpha^{(t)} \leftarrow \text{Aggregate}(\{\mathcal{R}_i^{(t)}\}_{i=1}^N)$
23:         **end for**
24:         /* Optimize graph generation parameters */
25:         $\Theta \leftarrow \Theta - \alpha \cdot \nabla_\Theta \mathcal{L}_{\text{CARD}}$
26:     **end for**
27: **end for**
---

## F PROMPT

---

**LLM Dynamic Information Template**

```
model_template = {
    'Name': '{model_name}',
    'Description': '{ModelName} is a {model_type} model developed
    ↪  by {developer}, supporting {modalities}. '
                   'It is optimized for {key_strengths}.  '
                   '{ModelName} offers {performance_advantage}. '
                   'The model costs ${input_cost} per million input
                   ↪  tokens and ${output_cost} per million output
                   ↪  tokens. '
                   '{evaluation_info}'
}

evaluation_info = (
    'In {domain_a}, {ModelName} achieves an accuracy of
    ↪  {evaluation_score_a}. '
    'In {domain_b}, {ModelName} achieves an accuracy of
    ↪  {evaluation_score_b}. '...
)
```

---

**Search Engine Dynamic Information Template**

```
search_engine_template = {
    'Name': '{engine_name}',
    'Description': '{EngineName} is a {engine_type} developed by
    ↪  {provider}. '
                   'It supports {supported_query_types} and
                   ↪  delivers results across {content_scope}. '
                   'The engine integrates {additional_features},
                   ↪  making it suitable for
                   ↪  {application_scenarios}. '
                   '{evaluation_info}'
}

evaluation_info = (
    'In the task of {task_name}, {EngineName} achieved a score of
    ↪  {score_value} on the {metric_name} metric. '
    ...
)
```

---

**HumanEval Role Profile**

```
"Project Manager":
    "You are a project manager. "
    "You will be given a function signature and its docstring by
    ↪  the user. "
    "You are responsible for overseeing the overall structure of
    ↪  the code, ensuring that the code is structured to complete
    ↪  the task Implement code concisely and correctly without
    ↪  pursuing over-engineering."
    "You need to suggest optimal design patterns to ensure that the
    ↪  code follows best practices for maintainability and
    ↪  flexibility. "
    "You can specify the overall design of the code, including the
    ↪  classes that need to be defined(maybe none) and the
    ↪  functions used (maybe only one function) ."
    "I hope your reply will be more concise. Preferably within
    ↪  fifty words. Don't list too many points.",
"Algorithm Designer":
    "You are an algorithm designer. "
    "You will be given a function signature and its docstring by
    ↪  the user. "
    "You need to specify the specific design of the algorithm,
    ↪  including the classes that may be defined and the functions
    ↪  used. "
    "You need to generate the detailed documentation, including
    ↪  explanations of the algorithm, usage instructions, and API
    ↪  references. "
    "When the implementation logic is complex, you can give the
    ↪  pseudocode logic of the main algorithm."
    "I hope your reply will be more concise. Preferably within
    ↪  fifty words. Don't list too many points.",
"Programming Expert":
    "You are a programming expert. "
    "You will be given a function signature and its docstring by
    ↪  the user. "
    "You may be able to get the output results of other agents.
    ↪  They may have passed internal tests, but they may not be
    ↪  completely correct. "
    "Write your full implementation (restate the function
    ↪  signature). "
    "Use a Python code block to write your response. For
    ↪  example:\n```python\nprint('Hello world!')\n```"
    "Do not include anything other than Python code blocks in your
    ↪  response. "
    "Do not change function names and input variable types in
    ↪  tasks.",
```

---

**HumanEval Role Profile**

```
"Test Analyst":
    "You are a test analyst. "
    "You will be given a function signature and its docstring by
    ↪  the user. "
    "You need to provide problems in the current code or solution
    ↪  based on the test data and possible test feedback in the
    ↪  question. "
    "You need to provide additional special use cases, boundary
    ↪  conditions, etc. that should be paid attention to when
    ↪  writing code. "
    "You can point out any potential errors in the code."
    "I hope your reply will be more concise. Preferably within
    ↪  fifty words. Don't list too many points.",
"Bug Fixer":
    "You are a bug fixer."
    "You will be given a function signature and its docstring by
    ↪  the user. "
    "You need to provide modified and improved python code based on
    ↪  the current overall code design, algorithm framework, code
    ↪  implementation or test problems. "
    "Write your full implementation (restate the function
    ↪  signature). "
    "Use a Python code block to write your response. For
    ↪  example:\n```python\nprint('Hello world!')\n```"
    "Do not include anything other than Python code blocks in your
    ↪  response "
    "Do not change function names and input variable types in
    ↪  tasks",
```

---

**MATH Role Profile**

```
"Math Solver":
    "You are a math expert. "
    "You will be given a math problem and hints from other agents.
    ↪    "
    "Give your own solving process step by step based on hints. "
    "The last line of your output contains only the final result
    ↪    without any units, for example: The answer is 140\n"
    "You will be given some examples you may refer to.",
"Mathematical Analyst":
    "You are a mathematical analyst. "
    "You will be given a math problem, analysis and code from other
    ↪    agents. "
    "You need to first analyze the problem-solving process step by
    ↪    step, where the variables are represented by letters. "
    "Then you substitute the values into the analysis process to
    ↪    perform calculations and get the results."
    "The last line of your output contains only the final result
    ↪    without any units, for example: The answer is 140\n"
    "You will be given some examples you may refer to.",
"Programming Expert":
    "You are a programming expert. "
    "You will be given a math problem, analysis and code from other
    ↪    agents. "
    "Integrate step-by-step reasoning and Python code to solve math
    ↪    problems. "
    "Analyze the question and write functions to solve the problem.
    ↪    "
    "The function should not take any arguments and use the final
    ↪    result as the return value. "
    "The last line of code calls the function you wrote and assigns
    ↪    the return value to the \(answer\) variable. "
    "Use a Python code block to write your response. For
    ↪    example:\n```python\ndef fun():\n x = 10\n y = 20\n return x
    ↪    + y\nanswer = fun()\n```\n"
    "Do not include anything other than Python code blocks in your
    ↪    response."
    "You will be given some examples you may refer to.",
"Inspector":
    "You are an Inspector. "
    "You will be given a math problem, analysis and code from other
    ↪    agents. "
    "Check whether the logic/calculation of the problem solving and
    ↪    analysis process is correct(if present). "
    "Check whether the code corresponds to the solution analysis(if
    ↪    present). "
    "Give your own solving process step by step based on hints. "
    "The last line of your output contains only the final result
    ↪    without any units, for example: The answer is 140\n"
    "You will be given some examples you may refer to.",
```

---

**MMLU Role Profile**

```
"Knowlegable Expert":
"""
You are a knowlegable expert in question answering.
Please give less than 3 key entities that need to be searched on
↪  the Internet to solve the problem. Each entity must be wrapped
↪  with @ symbols.
For example: @catfish effect@, @broken window effect@,
↪  @Shakespeare@.
If there is no entity in the question that needs to be searched on
↪  the Internet, you don't have to provide it.
"""
"Searcher":
"""
You will be given a question and Internet search overview of the
↪  key entities within it.
Please refer to them step by step to give your answer.
"""
"Critic":
"""
You are an excellent critic.
Please point out potential issues in other agent's analysis point
↪  by point.
"""
"Mathematician":
"""
You are a mathematician who is good at arithmetic calculation and
↪  long-term planning.
You can use your logic and reasoning skills to solve problems step
↪  by step.
"""
"Philosopher":
"""
You are a philosopher with deep knowledge in literature, history,
↪  and cultural studies.
You analyze texts critically, draw nuanced interpretations, and
↪  make connections across time, societies, and disciplines.
"""
"Doctor":
"""
You are a medical professional who good at biology, medicine, and
↪  health.
You combine modern medicine with herbal and natural remedies.
You consider age, lifestyle, and medical history in every
↪  recommendation.
"""
"Programmer":
"""
You are a programmer skilled in software development, systems
↪  design, and technical problem-solving.
You apply principles from computer science, engineering, and
↪  coding.
You write clean, efficient code across diverse platforms.
"""
```

Table 7: HumanEval Environment Configuration Set

|  | Configuration | LLM | Role |
|---|---|---|---|
| Train & Test | Configuration 1 | gpt-4o-mini
gpt-4o-mini
gpt-4o-mini
gpt-4o-mini
gpt-4o-mini | Project Manager
Algorithm Designer
Programming Expert
Test Analyst
Bug Fixer |
|  | Configuration 2 | deepseek-v3
deepseek-v3
deepseek-v3
deepseek-v3
deepseek-v3 | Project Manager
Algorithm Designer
Programming Expert
Test Analyst
Bug Fixer |
|  | Configuration 3 | llama-3-70B
llama-3-70B
llama-3-70B
llama-3-70B
llama-3-70B | Project Manager
Algorithm Designer
Programming Expert
Test Analyst
Bug Fixer |
| Only Test | Configuration 4 | gpt-4o
gpt-4o
gpt-4o
gpt-4o
gpt-4o | Project Manager
Algorithm Designer
Programming Expert
Test Analyst
Bug Fixer |
|  | Configuration 5 | qwen-2.5-72B
qwen-2.5-72B
qwen-2.5-72B
qwen-2.5-72B
qwen-2.5-72B | Project Manager
Algorithm Designer
Programming Expert
Test Analyst
Bug Fixer |

# G   EXPERIMENT CONFIGURATION SETS

Table 8: MATH Environment Configuration Set

|  | Configuration | LLM | Role |
|---|---|---|---|
| Train & Test | Configuration 1 | gpt-4o-mini
gpt-4o-mini
gpt-4o-mini
gpt-4o-mini
gpt-4o-mini | Math Solver
Mathematical Analyst
Mathematical Analyst
Programming Expert
Inspector |
|  | Configuration 2 | deepseek-v3
deepseek-v3
deepseek-v3
deepseek-v3
deepseek-v3 | Math Solver
Mathematical Analyst
Mathematical Analyst
Programming Expert
Inspector |
|  | Configuration 3 | llama-3-70B
llama-3-70B
llama-3-70B
llama-3-70B
llama-3-70B | Math Solver
Mathematical Analyst
Mathematical Analyst
Programming Expert
Inspector |
| Only Test | Configuration 4 | gpt-4o
gpt-4o
gpt-4o
gpt-4o
gpt-4o | Math Solver
Mathematical Analyst
Mathematical Analyst
Programming Expert
Inspector |
|  | Configuration 5 | qwen-2.5-72B
qwen-2.5-72B
qwen-2.5-72B
qwen-2.5-72B
qwen-2.5-72B | Math Solver
Mathematical Analyst
Mathematical Analyst
Programming Expert
Inspector |

Table 9: MMLU Environment Configuration Set

|  | Configuration | LLM | Role |
|---|---|---|---|
| Train & Test | Configuration 1 | gpt-4o-mini
gpt-4o-mini
gpt-4o-mini
gpt-4o-mini
gpt-4o-mini | Mathematician
Programmer
Critic
Doctor
Psychologist |
|  | Configuration 2 | deepseek-v3
deepseek-v3
deepseek-v3
deepseek-v3
deepseek-v3 | Mathematician
Programmer
Critic
Doctor
Psychologist |
|  | Configuration 3 | llama-3-70B
llama-3-70B
llama-3-70B
llama-3-70B
llama-3-70B | Mathematician
Programmer
Critic
Doctor
Psychologist |
| Only Test | Configuration 4 | gpt-4o
gpt-4o
gpt-4o
gpt-4o
gpt-4o | Mathematician
Programmer
Critic
Doctor
Psychologist |
|  | Configuration 5 | qwen-2.5-72B
qwen-2.5-72B
qwen-2.5-72B
qwen-2.5-72B
qwen-2.5-72B | Mathematician
Programmer
Critic
Doctor
Psychologist |

Table 10: MMLU Environment Configuration Set with External Tools

|  | Configuration | LLM | Role | External Tool (search engine) |
|---|---|---|---|---|
| Train & Test | Configuration 1 | gpt-4o-mini
gpt-4o-mini
gpt-4o-mini
gpt-4o-mini
gpt-4o-mini | Knowlegable Expert
Searcher
Psychologist
Mathematician
Critic | Google |
| Only Test | Configuration 2 | gpt-4o
gpt-4o
gpt-4o
gpt-4o
gpt-4o | Knowlegable Expert
Searcher
Psychologist
Mathematician
Critic | Google |
|  | Configuration 3 | gpt-4o-mini
gpt-4o-mini
gpt-4o-mini
gpt-4o-mini
gpt-4o-mini | Knowlegable Expert
Searcher
Psychologist
Mathematician
Critic | Wiki |
| Train & Test | Configuration 4 | llama-3-70B
llama-3-70B
llama-3-70B
llama-3-70B
llama-3-70B | Knowlegable Expert
Searcher
Psychologist
Mathematician
Critic | Wiki |

