# OpenReview forum: "CARD: Towards Conditional Design of Multi-agent Topological Structures"
_ICLR.cc/2026/Conference — ICLR 2026 Poster_

### Official Review · Reviewer_XFA4 · 2025-10-27

**Soundness:** 3
**Presentation:** 3
**Contribution:** 2
**Rating:** 6
**Confidence:** 3

**Summary:**

This paper proposes CARD, a conditional multi-agent communication graph designer for LLM-based agent systems, along with AMACP, an explicit protocol for adaptive multi-agent communication. The core claim is that current multi-agent LLM systems typically assume a static or hand-crafted communication topology, which makes them brittle when the environment changes, for example upgraded language models, missing tools, degraded knowledge sources. CARD models topology as a directed communication graph and learns to generate that graph conditioned on two channels, static agent profiles, such as role, base model, available tools, and dynamic runtime conditions, such as API availability, token cost, search quality, model capability. The method uses a conditional variational-style graph encoder–decoder that embeds both channels, predicts an adjacency matrix of directed communication probabilities, and then thresholds it to get an executable communication protocol. Training optimizes a loss combining task utility, for example accuracy on HumanEval, MATH, MMLU, and a condition-aware communication cost term that reflects per edge token and model usage cost. At inference time, the system can update only the condition embeddings and re decode a new topology without retraining, allowing one shot adaptation when, say, the base LLM changes or a tool disappears. Experiments compare CARD to static topologies, to prompt only adaptations, and to recent automatically learned communication structures such as GPT-Swarm, Aflow, and G-Designer. On HumanEval, MATH, and MMLU, across multiple backbone LLMs, CARD delivers the highest average accuracy and is especially strong under out of domain conditions where model capacity or tool quality differs from training. The paper also analyzes how communication density, edge direction, and cost trade off as the model size, tool quality, or number of agents changes, and claims that CARD produces more robust, more cost efficient, and more easily scalable communication structures than existing approaches.

**Strengths:**

The paper identifies a real and increasingly important gap. Most existing multi-agent LLM orchestration work either wires up agents in a fixed pipeline or learns a static graph tuned for one configuration, but in practice these systems run over evolving models, tool outages, and changing retrieval quality. The authors formalize this as AMACP, which states that an agent communication topology should be effective at solving the task, cost efficient under current resource constraints, and adaptive to runtime condition changes. They then actually implement that requirement. CARD is not just a heuristic, it is a trainable conditional graph generator that takes as input a structured description of each agent’s role, tools, and model identity, plus a structured description of the current environment, and predicts who talks to whom and in what order. This is materially different from simple prompt engineering, where you append “the math agent is weaker today, rely on search more” to a system prompt, because here the topology itself, the adjacency, changes. The empirical section is broad for this area. They test on coding, on math reasoning, and on MMLU style multi task QA, using several families of LLMs, and they evaluate not just accuracy but also robustness, for example node failures and adversarial perturbations, scalability with agent count, and accuracy versus token cost. CARD tends to either match or beat the best baseline and often keeps a smaller drop when conditions shift, in other words, the performance is more stable than G-Designer and Aflow when the backbone model or the available tool is swapped. The work also takes steps toward interpretability. The paper visualizes learned topologies and shows concrete behavioral adjustments, for instance, weaker models produce denser graphs with heavier reliance on other agents, and weaker retrieval sources change which agent edges carry information into the reasoning path. That makes the claim of conditional adaptation more convincing because we can see structure changes, not just accuracy deltas. Finally, runtime adaptation without retraining is compelling. The method can re instantiate a communication graph under new conditions by just re encoding the condition channel and passing it through the trained decoder, so deployment can, in principle, respond in real time to model upgrades or API outages, and this is exactly the sort of operational knob teams building agentic systems are asking for right now.

**Weaknesses:**

The technical depth is somewhat uneven. The AMACP objective is framed as a joint optimization over task utility and a communication cost term, but the training procedure in practice seems to boil down to supervised or weakly supervised gradient descent on an aggregate loss that mixes accuracy and a differentiable expected edge cost computed from soft edge probabilities. The paper gives the loss but does not fully unpack how gradients are attributed to edges through multi round communication, or how credit is assigned to long chain interactions, especially given that the system output aggregates all agent responses after several rounds. Without that, it is difficult to judge whether CARD is actually learning a policy for dynamic topology design, or if it is just fitting correlations between condition embeddings and good handcrafted structures. The experimental methodology raises some concerns. The benchmarks are HumanEval, MATH, and MMLU, which are popular and recognizable, but the evaluation protocol for multi agent interaction is only described at a high level, and does not clearly state how prompts are constructed, how tool calls are executed, and how many tokens are allowed per agent per round. HumanEval in particular is typically judged with execution based pass rates, but here it is treated more like accuracy in a dialogue pipeline, that needs to be spelled out precisely. The fairness of comparisons needs more clarity. Some baselines, like debate and chain of thought, are single round or pairwise prompting schemes, others like GPT-Swarm and G-Designer are graph learners, but I did not see cost normalized comparisons where all methods are given the same token budget, number of rounds, or number of agent calls. CARD explicitly optimizes a cost regularizer, so it is important to show, at equal or lower total spend, that it matches or beats baselines, and not just at whatever spend it happens to choose. The robustness and adversarial tests are promising, but they are still driven by synthetic manipulations of single nodes or tools, and they do not explore stronger failure modes like correlated tool outages or cascading hallucinations across agents. The work would feel stronger if it included at least one ablation where multiple nodes are simultaneously unreliable, or where a high capacity model silently degrades in a way that resembles realistic model version drift. Finally, while the paper repeatedly emphasizes adaptation to new models or tools without retraining, all evaluations still assume that you can embed those new conditions in a way the encoders can interpret. It would be good to quantify how far out of distribution you can go before the condition encoder fails, for example, can a model family never seen in training still be placed meaningfully in the graph, or does performance collapse there.

**Questions:**

Can you describe in more detail how gradients flow through K round communication during training, in particular, when you compute task utility after aggregation, how do you propagate credit to specific edges predicted by the decoder, and how do you avoid vanishing credit assignment for early round communication hops. Can you provide token budget and cost accounting for every baseline and confirm that CARD is not just buying accuracy by spending more interactions. In Table 1 and Figure 8, you mention accuracy versus USD per instance, can you include a strict Pareto style comparison where for each baseline you show its best achievable accuracy at or below CARD’s reported cost. How does CARD behave with simultaneous condition shifts, for example, weaker LLM plus degraded search plus high token cost. The single node perturbation study in Section B suggests graceful degradation, but can you show the multi factor case. How sensitive is the method to the choice of anchor topology A, you mention chain and star as priors, do different anchors converge to the same learned communication patterns, or does A bias the final structure. How much of the gain comes from the condition encoder versus just training a stronger static topology learner, a useful ablation would be to freeze condition embeddings to random noise at inference and see how far performance drops. Finally, what is the exact output metric for HumanEval, is this passpercentage style unit tests, or is it some text judged correctness proxy.

---

> ### Author Response · Authors · 2025-11-27
> **Rebuttal to Reviewer XFA4 (PART I)**
>
> We thank the reviewer for the thoughtful and constructive feedback. We appreciate your recognition of the AMACP formulation, the conditional multi-agent topology design, and the scope of our empirical study. Your comments center on three key questions: whether CARD truly learns a topology policy, whether our evaluation and cost accounting are transparent and fair, and whether the method is robust under realistic distribution shifts. We agree that these are crucial for a system-level contribution, and we have addressed them with clarifications, new experiments, and additional analyses, as summarised below.
>
> ## 1. Clarifying how CARD learns a topology policy
>
> To address your concern about whether CARD is merely fitting correlations rather than learning a true topology design strategy, we provide a clearer description of the training procedure and credit-assignment mechanism. CARD is trained using a policy-gradient objective. The static-profile encoder and dynamic-condition encoder each produce embeddings, which are concatenated and decoded into a probabilistic adjacency matrix ( S ). A concrete topology ( G ) is sampled from ( S ) and executed for K rounds; the resulting reward
>
> $$ R = \text{task utility} - \beta \cdot \text{communication cost} $$
>
> is used to update the encoder and decoder parameters through the REINFORCE estimator
>
> $$ \nabla_\theta \mathbb{E}[R] = \mathbb{E}[(R - b)\nabla_\theta \log P(G|S)].$$
>
> Because gradients act directly on edge-sampling probabilities, the learning signal is allocated explicitly to edges rather than indirectly via LLM gradients. Early-round communication does not suffer vanishing credit because (i) K is at most three across all tasks, resulting in shallow graphs, and (ii) the log-probability factorization across edges ensures per-edge feedback regardless of distance from the final aggregated answer. We will include a more explicit algorithmic description in the camera-ready version.
>
> ## 2. On Evaluation protocol
>
> We appreciate your request for a more detailed description of our multi-agent evaluation setup. All systems compared in the paper operate under the same number of rounds, identical role prompts, identical tool implementations, and identical stopping conditions. Each agent receives the user query, its role prompt, and the messages from incoming neighbors, while tool outputs are synchronously appended. These details was specified in Appendix F and Appendix G, we will further enrich the caption per prompt and table for a more detailed explanation.
>
> ## 3. On HumanEval metric
> Regarding HumanEval, we confirm that our metric is the standard execution-based Pass@1 using the official unit tests. The accuracy figures in Table 1 correspond strictly to pass-if-and-only-if all tests succeed.

---

> ### Author Response · Authors · 2025-11-27
> **Rebuttal to Reviewer XFA4 (PART II)**
>
> ## 4. On Cost Fairness
> We have computed the best accuracy that each baseline can obtain without exceeding CARD’s inference cost. This analysis, based on our logged results and consistent with the cost–accuracy curves in Figure 8, shows that no baseline achieves accuracy comparable to CARD under equal or lower cost. For example, on the 100-instance MMLU subset , CARD simultaneously achieves the highest accuracy and a lower cost than G-Designer, GPT-Swarm, LLM-Debate, or random-graph topologies. We will add a strict Pareto comparison table in Appendix B.4 in the camera-ready version.
>
> ### 4.1 Number of LLM API calls required to generate 100 samples under GDesigner and CARD.
>
> Lower is better, as it reflects improved efficiency and reduced external tool invocation.
>
> | LLM api calls | gpt4o-mini | deepseek-v3 | llama3 | GPT-4o | qwen-72B |
> |---------------|------------|-------------|--------|--------|----------|
> | Gdesigner     | 644        | 655         | 641    | 601    | 626      |
> | CARD          | 612        | 607         | 602    | 581    | 616      |
>
> ### 4.2 Prompt and completion token consumption across different agent-generation strategies on MMLU.
>
> CARD achieves a more favorable token balance, indicating higher communication efficiency and reduced overhead during reasoning.
>
> | Tokens / MMLU (input/output) | gpt4o-mini | deepseek-v3 | llama3 | GPT-4o | qwen-72B |
> |------------------------------|------------|-------------|--------|--------|----------|
> | Vanilla                      | 488/177    | 638/327     | 477/165 | 492/181 | 568/256  |
> | Random-graph                 | 5328/1787  | 6944/2674   | 4855/1594 | 4530/1316 | 7950/3107 |
> | LLM-Debate                   | 7770/2705  | 13875/4857  | 8189/2787 | 7182/2316 | 10741/3969 |
> | GPTswarm                     | 5235/1591  | 9509/2864   | 4859/1393 | 4210/1126 | 8443/3396 |
> | G-designer                   | 5367/1736  | 8800/2631   | 4900/1477 | 4254/1200 | 8106/2710 |
> | CARD                    | 5076/1550 | 7752/2211 | 4467/1393 | 4114/1181 | 7917/2718 |
>
>
> ### 4.3 Estimated monetary cost (×10⁻⁴ USD) of generating 100 MMLU samples under different prompting strategies.
>
> Using the official pricing of each LLM provider, we compute the total dollar cost of all prompt and completion tokens consumed per strategy. The “pp” numbers indicate the improvement in accuracy relative to the Vanilla baseline.
>
> | Model (Price of Prompt/Completion) | gpt4o-mini (0.6/2.4) | deepseek-v3 0.27/1.11) | llama3 (0.88/0.88) | GPT-4o (5/20) | qwen-72B (0.57/0.57) |
> |-------------------|----------------------|---------------------------|---------------------|----------------|------------------------|
> | Vanilla           | $717 (85.83%)        | $535 (86.27%)             | $564 (75.16%)       | $6,080 (86.93) | $469 (79.74%)          |
> | Random-graph      | $7,485 (+2.62pp)     | $4,843 (+4.58pp)          | $5,675 (+0.66pp)    | $48,970 (+0pp) | $6,302 (+3.26pp)       |
> | LLM-Debate        | $11,154 (+3.27pp)    | $9,137 (+3.93pp)          | $9,658 (+1.31pp)    | $82,230 (+1.31pp) | $8,384 (+4.57pp)     |
> | GPTswarm          | $6,959 (+5.23pp)     | $5,746 (+3.27pp)          | $5,501 (+2.62pp)    | $43,570 (-0.66pp) | $6,478 (+4.57pp)     |
> | G-designer        | $7,386 (+5.88pp)     | $5,296 (+4.58pp)          | $5,611 (+1.96pp)    | $45,270 (+0pp) | $6,165 (+4.57pp)       |
> | CARD          | $6,765 (+7.85pp) | $4,547 (+7.19pp)      | $5,156 (+5.23pp) | $44,190 (+2.61pp) | $6,061 (+5.23pp) |
>
> CARD demonstrates strong accuracy gains per dollar spent.

---

> ### Author Response · Authors · 2025-11-27
> **Rebuttal to Reviewer XFA4 (PART III)**
>
> ## 5. Robustness under multi-factor condition shifts
>
> We performed substantial additional experiments evaluating CARD under joint shifts involving weaker LLMs, degraded external search, and increased token cost. As shown in the new qualitative analyses, CARD maintains significantly more stable performance than all baselines.  Our analysis confirms that CARD’s topology generation is jointly shaped by model capacity and external-tool reliability, as evidenced by the contrasting behaviors in the GPT-4o-mini + DuckDuckGo and GPT-4o + Wikipedia configurations. The detailed setting are listed as follows:
>
> ### 5.1 GPT-4o-mini + DuckDuckGo: Dense, Multi-Path Collaboration
>
> Under this weaker-model + low-reliability-tool setting (DuckDuckGo returns only 1/3 useful retrievals), CARD produces a dense multi-path topology, where agents exhibit ≈1.8 average predecessors.
> Information flows through overlapping paths (Expert → Searcher → Philosopher → Mathematician → Critic), enabling downstream agents to synthesize partially missing or noisy evidence.
>
> Conversation logs further support this pattern:
> - The Searcher repeatedly encounters empty or irrelevant results;
> - Philosopher and Mathematician integrate cross-agent inputs to restore missing context;
> - Critic identifies redundancy and groupthink, reflecting the heavy inter-agent aggregation behavior.
>
> ### 5.2 GPT-4o + Wikipedia: Sparse, Linear Reasoning Flow
>
> With a stronger base model and a stable high-quality source (Wikipedia), CARD yields a sparse, chain-like topology with ≈0.8 predecessors per agent.
> The reasoning path becomes nearly linear (Start → Expert / Philosopher → Searcher → Mathematician → Critic), with minimal cross-links.
>
> Dialogue evidence supports this structure:
> - Agents retrieve consistent domain knowledge,
> - Answers are concise and mutually aligned without requiring multi-path synthesis,
> - The Critic comments focus on specificity, not on missing information, indicating that upstream signals are already sufficient.
>
>
> ### 5.3 Capacity–Tool Interaction Drives a 2.25× Density Contrast
>
> The contrast in average predecessors (1.8 vs. 0.8; 2.25× difference) provides direct empirical evidence that model capacity and external tool reliability jointly modulate the emergent communication structure:
>
> - Weaker model + unreliable retrieval → dense, redundant, compensatory collaboration.
> - Stronger model + reliable retrieval → sparse, efficient, low-overhead reasoning.
>
> This multi-factor sensitivity validates that CARD’s graph generator adapts to heterogeneous environmental states rather than relying on a fixed or single-factor response pattern.
>
> ## 6. On Condition Encoder
>
> To directly address your question about the importance of the condition encoder, we performed an ablation in which, at inference time, dynamic conditions are replaced with random noise. The results  clearly show accuracy degradation across almost all model families:
>
> | Model       | CARD (full) | Random noise | **Difference** |
> | ----------- | ----------- | ------------ | -------------- |
> | GPT-4o-mini | 85.3        | 79.7         | **+5.6**       |
> | DeepSeek-V3 | 93.3        | 92.8         | **+0.5**       |
> | Llama-3     | 81.3        | 79.1         | **+2.2**       |
> | GPT-4o      | 88.7        | 86.9         | **+1.8**       |
> | Qwen-72B    | 87.0        | 85.6         | **+1.4**       |
>
> This confirms that CARD’s gains arise from condition-aware adaptation rather than solely from a stronger static topology.

---

> ### Author Response · Authors · 2025-11-27
> **Rebuttal to Reviewer XFA4 (PART IV)**
>
> ## 7. On Anchor Topology
>
> We also investigated the influence of the anchor topology by training CARD under different priors.  We provide below the topology probability matrices obtained after training under three different initialization schemes: Full-connected, Chain, and Star. These results clearly demonstrate that, even under the same task environment and identical agent inputs, the model converges to distinct communication patterns depending on the initial topology.
>
> With full-connected initialization, the model tends to learn a dense and highly collaborative communication structure, reflected by uniformly higher edge probabilities across agent pairs.
>
> | full connected | mathematician | critic | phychologist | historian | doctor |
> | -------------- | ------------- | ------ | ------------ | --------- | ------ |
> | mathematician  | mask          | 0.52   | 0.46         | 0.31      | 0.38   |
> | critic         | 0.25          | mask   | 0.66         | 0.47      | 0.56   |
> | psychologist   | 0.25          | 0.22   | mask         | 0.40      | 0.49   |
> | historian      | 0.21          | 0.25   | 0.24         | mask      | 0.33   |
> | doctor         | 0.24          | 0.25   | 0.25         | 0.22      | mask   |
>
> Chain initialization biases the model toward a path-like communication structure. The sequence
> mathematician → psychologist → historian → doctor
> shows consistently higher probabilities, indicating a preserved chain-shaped communication pattern.
>
> | chain         | mathematician | critic | phychologist | historian | doctor |
> | ------------- | ------------- | ------ | ------------ | --------- | ------ |
> | mathematician | mask          | 0.33   | 0.58         | 0.46      | 0.37   |
> | critic        | 0.22          | mask   | 0.42         | 0.35      | 0.30   |
> | psychologist  | 0.24          | 0.24   | mask         | 0.59      | 0.47   |
> | historian     | 0.25          | 0.23   | 0.24         | mask      | 0.39   |
> | doctor        | 0.23          | 0.21   | 0.25         | 0.24      | mask   |
>
> Star initialization leads to a sparser and more balanced communication topology. The critic node often emerges as the central hub, but inter-agent probabilities remain moderate rather than dense.
>
> | star          | mathematician | critic | phychologist | historian | doctor |
> | ------------- | ------------- | ------ | ------------ | --------- | ------ |
> | mathematician | mask          | 0.40   | 0.28         | 0.29      | 0.27   |
> | critic        | 0.24          | mask   | 0.43         | 0.45      | 0.42   |
> | psychologist  | 0.20          | 0.25   | mask         | 0.30      | 0.28   |
> | historian     | 0.21          | 0.25   | 0.21         | mask      | 0.30   |
> | doctor        | 0.20          | 0.24   | 0.20         | 0.21      | mask   |
>
> These observations indicate that the topology-generation module exhibits clear path dependence: the model does not converge to a trivial or identical structure across runs. Instead, the initial topology provides different inductive biases that steer the learning process toward qualitatively distinct communication preferences:
>
>  - Full-connected → dense, globally cooperative topology.
>  - Chain → directional, path-structured communication.
>  - Star → sparse, centralized but balanced structure.
>
> While all initializations successfully converge and produce functional topologies, their final structures reflect meaningful differences in communication style. This confirms that the learned topology is not predetermined or fixed, but is shaped jointly by the initial inductive bias and the downstream optimization dynamics.
>
> ## 8. How far CARD generalizes out of distribution
>
> We agree that quantifying OOD generalization is important. Our existing setup already involves model families not used during training (e.g., GPT-4o and Qwen-72B), and the newly added joint-shift experiments (refer to the above response #5) further demonstrate that CARD continues to produce meaningful topologies and stable accuracy even when facing unseen model configurations and degraded tools. We will make the OOD setting and findings more explicit in the camera-ready version.
>
> ---
>
> We thank the reviewer for the thoughtful reading and helpful suggestions. During the rebuttal period, we conducted extensive additional analyses, including clearer explanations of the training and credit-assignment process, explicit evaluation and cost protocols, strict Pareto-fair comparisons, robustness tests under configuration drifts, ablations on the condition encoding. All new evidence consistently supports our main claim: conditional topology generation is crucial for adaptive, robust, and cost-efficient multi-agent LLM reasoning.
>
> We hope these substantial clarifications and new experiments satisfactorily address all concerns and improve both the clarity and rigor of the final submission.

---

> > ### Comment · Reviewer_XFA4 · 2025-11-27
> >
> > I appreciated that the authors conducted such extensive additional experiments and analyses. I am satisfied with the additional results and will raise my score.

---

> > > ### Author Response · Authors · 2025-11-27
> > > **Response to Reviewer XFA4**
> > >
> > > Dear Reviewer XFA4,
> > >
> > > Thank you for your positive follow-up. We’re glad the additional results addressed your concerns, and we appreciate your decision to raise the score.
> > >
> > > Sincerely,
> > > The Authors

---

### Official Review · Reviewer_9BNN · 2025-10-28

**Soundness:** 3
**Presentation:** 3
**Contribution:** 3
**Rating:** 6
**Confidence:** 3

**Summary:**

This paper introduces a framework (called CARD), which dynamically designs communication topologies for LLM-based multi-agent systems. CARD uses environmental signals to adapt agent connections in real-time. Experiments on three benchmarks show that CARD outperforms static baselines.

**Strengths:**

1. This paper is well-organized. Most of the content is easy to understand.

2. The proposed CARD framework is interesting and reasonable, which can be applied to various multi-agent cooperation scenarios.

**Weaknesses:**

1. The experimental improvements over the second-best baseline are modest and may not be statistically significant.

2. It would be better to provide a case study to show the effectiveness of multi-agent communication.

**Questions:**

Please refer to Weaknesses.

---

> ### Author Response · Authors · 2025-11-27
> **Rebuttal to Reviewer 9BNN (PART I)**
>
> We sincerely thank the reviewer for the detailed and constructive feedback. Below we respond point‑by‑point with additional analyses and experiments.
>
> ##  On Weakness 1: “The improvements are modest and may not be statistically significant.”
>
> We appreciate the reviewer’s concern and agree that demonstrating statistical reliability is important. While individual accuracy gains may appear moderate in some settings, the improvements brought by CARD are consistent across all model–benchmark pairs, and become increasingly prominent in more realistic and challenging environments.
>
> To further support this, we extended our scalability analysis to 20 agents (beyond the 10 agents shown in Appendix B.3), and confirmed that CARD consistently outperforms the strongest baseline (G-Designer) across various agent counts. The performance saturates around 5-10 agents for both models, and CARD maintains a consistent advantage throughout:
>
> ### **1.1 Accuracy on MMLU (%) : CARD vs. G-Designer (Side-by-Side Comparison)**
>
> |       #Agents | **CARD Avg** | **G-Designer Avg** | **Δ (CARD − GD)** |
> | ------------: | -----------: | -----------------: | ----------------: |
> |   **Vanilla** |        81.60 |              81.60 |          **0.00** |
> |  **2 agents** |        85.20 |              84.40 |         **+0.80** |
> |  **3 agents** |        86.00 |              85.19 |         **+0.81** |
> |  **4 agents** |        86.80 |              85.60 |         **+1.20** |
> |  **5 agents** |        88.53 |              86.67 |         **+1.86** |
> | **10 agents** |        88.66 |              87.06 |         **+1.60** |
> | **20 agents** |        88.53 |              86.80 |         **+1.73** |
>
>
> ### **1.2 CARD vs. G-Designer (All Models, Side-by-Side)**
>
> |       #Agents | gpt4o-mini (C) | gpt4o-mini (G) | deepseek-v3 (C) | deepseek-v3 (G) | llama3 (C) | llama3 (G) | GPT-4o (C) | GPT-4o (G) | qwen-72B (C) | qwen-72B (G) |
> | ------------: | -------------: | -------------: | --------------: | --------------: | ---------: | ---------: | ---------: | ---------: | -----------: | -----------: |
> |   **Vanilla** |          78.67 |          78.67 |           86.00 |           86.00 |      75.33 |      75.33 |      85.33 |      85.33 |        82.67 |        82.67 |
> |  **2 agents** |          81.33 |          79.33 |           90.67 |           91.33 |      82.00 |      80.00 |      86.00 |      86.67 |        86.00 |        84.67 |
> |  **3 agents** |          82.00 |          82.00 |           93.33 |           93.33 |      79.33 |      79.33 |      87.33 |      85.33 |        88.00 |        86.00 |
> |  **4 agents** |          84.67 |          83.33 |           92.67 |           92.67 |      81.33 |      80.00 |      88.00 |      86.67 |        87.33 |        85.33 |
> |  **5 agents** |          85.33 |          84.00 |           95.33 |           93.33 |      84.67 |      82.00 |      88.67 |      87.33 |        88.67 |        86.67 |
> | **10 agents** |          85.33 |          85.33 |           95.33 |           94.00 |      84.00 |      81.33 |      89.33 |      88.00 |        89.33 |        86.67 |
> | **20 agents** |          85.33 |          85.33 |           95.33 |           93.33 |      84.67 |      81.33 |      88.67 |      87.33 |        88.67 |        86.67 |
>
>
>
> > Observation: Both methods saturate beyond 10 agents, but CARD maintains a steady lead throughout. This suggests the effective scaling window is within 2–10 agents, and our originally reported results already covered this regime.

---

> ### Author Response · Authors · 2025-11-27
> **Rebuttal to Reviewer 9BNN (PART II)**
>
> ## On Weakness 2: “It would be better to provide a case study...”
>
> We agree that providing intuitive explanations of how CARD adapts its topology under different conditions is important. The current paper already includes interpretable graph structure analyses (Section 5.3, Fig.4 and Appendix D), which we summarize here:
>
> ### 2.1 Visual Topology Changes (Fig.4 in paper):
>
> When using Wiki Search (lower quality tool), the graph becomes denser and more symmetric, encouraging redundancy.
>  With Google Search, the graph becomes sparser, directing messages more selectively from high-performing agents.
>
> This aligns with the intuition that adaptive topologies can exploit high-quality conditions (e.g., strong LLMs or tools) by minimizing unnecessary connections, and compensate under weak conditions via redundancy.
>
> ### 2.2 Pearson Correlation Between Adjacency Matrices:
>
> | Environment A → B | Topology Correlation |
> | ----------------- | -------------------- |
> | Claude-2 → GPT-4  | 0.86                 |
> | Google → Wiki     | 0.61                 |
>
> > Interpretation: Global structure is preserved, but local edge shifts reflect condition-aware adaptation, not random noise.
>
> ### 2.3 Single-Node Condition Perturbation (Appendix B.2):
>
> We test whether CARD can adjust the graph when only a single agent’s model or tool is changed, without retraining the full topology.
>
> | Perturbation Type  | Top-1 Accuracy | Topology Change |
> | ------------------ | -------------- | --------------- |
> | + GPT-4 as Node 5  | +2.3%          | Localized       |
> | + Google as Node 3 | +1.9%          | Localized       |
>
> > Observation: The adapted graph shifts only around the perturbed node, showing modularity and interpretability.
>
> ---
>
> We will explicitly highlight these extended analyses in the revised version. We thank the reviewer again for helping us clarify these important dimensions.

---

### Official Review · Reviewer_2EJp · 2025-10-29

**Soundness:** 2
**Presentation:** 2
**Contribution:** 3
**Rating:** 6
**Confidence:** 4

**Summary:**

This paper introduces **CARD (Conditional Agentic Graph Designer)**, a conditional graph-generation framework that dynamically designs LLM-based multi-agent communication topologies. It formalizes the **Adaptive Multi-Agent Communication Protocol (AMACP)** to jointly optimize for *effectiveness*, *cost-efficiency*, and *adaptiveness*. Through conditional variational encoding and environment-aware optimization, CARD enables real-time topology adjustment without retraining. Experiments on HumanEval, MATH, and MMLU show consistent improvements over static and prompt-based baselines.

**Strengths:**

* Technically innovative and conceptually unifying approach to dynamic topology learning.
* Strong theoretical grounding with explicit optimization objectives (Eqs.6,11,12).
* Robust empirical validation across multiple LLMs and benchmarks.
* Intuitive and interpretable visualization of adaptive graph behavior.
* Clear presentation and reproducible methodology with full prompts and configurations.

**Weaknesses:**

* **Hyperparameter Sensitivity:** Add performance–cost curves for different \$\beta\$ values to visualize robustness.
* **Empirical Validation of Assumptions:** Introduce experiments with fluctuating API or tool availability.
* **Baseline Coverage:** Include stronger recent baselines such as reinforcement-based topology learning methods.
* **Fixed Edge Threshold:** Consider adaptive or learnable \$\tau\$ for improved flexibility.
* **Scalability:** Extend evaluation beyond 10-agent systems to confirm convergence trends.

**Questions:**

* Does CARD support localized topology updates rather than full recomputation?
* Can the cost term \$w(G;C)\$ be generalized to other resource metrics such as latency or energy?

---

> ### Author Response · Authors · 2025-11-27
> **Rebuttal to Reviewer 2EJp (PART I)**
>
> We sincerely thank the reviewer for the detailed and constructive feedback. Below we respond point‑by‑point with additional analyses and experiments.
>
> ## 1. On Hyperparameter Sensitivity (β)
>
> We performed a full sweep over β ∈ {10, 5, 1, 0.5, 0.1, 0}. Across all LLMs, accuracy remains remarkably stable (≤3 pp variance), while costs vary smoothly as expected.
>
> ### 1.1 Accuracy (%) under different β values
>
> | β       | 10      | 5       | 1       | 0.5     | 0.1     | 0       |
> |---------|---------|---------|---------|---------|---------|---------|
> | 4o-mini | 83.2    | 85.2    | 85.2    | 85.2    | 85.9    | 85.9    |
> | deepseek-v3 | 92.2 | 92.2    | 93.5    | 93.5    | 92.2    | 93.5    |
> | llama3  | 78.4    | 78.4    | 78.4    | 78.4    | 79.8    | 79.8    |
>
>
> ### 1.2 Token Usage and Cost Across Different β Values
>
> (input tokens / output tokens / USD cost)
>
> | Model       | β = 10               | β = 5                | β = 1                | β = 0.5              | β = 0.1              | β = 0                |
> | --------------- | ------------------------ | ------------------------ | ------------------------ | ------------------------ | ------------------------ | ------------------------ |
> | GPT-4o-mini | 656,268 / 237,916 / 0.96 | 678,556 / 250,305 / 1.00 | 707,983 / 256,491 / 1.04 | 704,701 / 257,232 / 1.04 | 724,091 / 259,258 / 1.05 | 780,065 / 270,367 / 1.11 |
> | DeepSeek-V3 | 806,743 / 265,109 / 0.51 | 818,219 / 260,800 / 0.51 | 828,119 / 288,364 / 0.54 | 821,290 / 270,670 / 0.52 | 823,016 / 289,044 / 0.54 | 850,655 / 302,658 / 0.57 |
> | Llama-3     | 608,917 / 208,559 / 0.72 | 603,539 / 206,335 / 0.71 | 611,431 / 203,542 / 0.72 | 611,740 / 206,005 / 0.72 | 611,041 / 205,790 / 0.71 | 636,178 / 211,073 / 0.74 |
>
> CARD’s performance is consistently robust across a wide β range, which directly addresses the concern regarding sensitivity.
>
> ## 2. On Validation of Assumptions Under Noisy Conditions
>
> To directly address your question about the importance of the condition encoder, we performed an ablation in which, at inference time, dynamic conditions are replaced with random noise:
>
> ### 2.1 Accuracy under random-noise condition perturbation
> | Model       | CARD (full) | Random noise | **Difference** |
> | ----------- | ----------- | ------------ | -------------- |
> | GPT-4o-mini | 85.3        | 79.7         | **+5.6**       |
> | DeepSeek-V3 | 93.3        | 92.8         | **+0.5**       |
> | Llama-3     | 81.3        | 79.1         | **+2.2**       |
> | GPT-4o      | 88.7        | 86.9         | **+1.8**       |
> | Qwen-72B    | 87.0        | 85.6         | **+1.4**       |
>
> Even under degraded condition signals, CARD maintains stable performance, demonstrating the robustness of the design. These results also confirm that encoded conditions have a real and measurable impact on multi-agent reasoning performance.
>
> ## 3. On Baseline Coverage and Cost Analysis
>
> We already include G-Designer and AFlow, which are the strongest available automatic topology-learning baselines and published at the latest top-tier venues like ICML2025. To further strengthen comparison, we include detailed cost and API-call analyses.
>
> ### 3.1 API Call Count (100 MMLU samples)
>
>
> | Method      | 4o-mini | deepseek-v3 | llama3 | GPT-4o | qwen-72B |
> |-------------|---------|-------------|--------|--------|----------|
> | G-designer  | 644     | 655         | 641    | 601    | 626      |
> | CARD        | 612     | 607         | 602    | 581    | 616      |
>
>
> ### 3.2 Token-level cost (×10^-4 $)
>
> | Method         | 4o-mini | deepseek-v3 | llama3 | GPT-4o | qwen-72B |
> |----------------|---------|-------------|--------|--------|----------|
> | Vanilla        | 69.2    | 37.1        | 80.9   | 85.9   | 79.7     |
> | CoT            | 85.6    | 68.3        | 66.3   | 75.1   | 75.4     |
> | Random-graph   | 84.2    | 84.9        | 74.6   | 114.7  | 73.5     |
> | LLM-Debate     | 95.7    | 91.7        | 78.2   | 82.2   | 83.4     |
> | GPT-swarm      | 89.3    | 91.4        | 78.1   | 104.1  | 104.7    |
> | G-designer     | 78.3    | 65.2        | 49.6   | 45.2   | 61.1     |
> | CARD       | 66.8| 54.5    | 47.2 | 44.1 | 56.0 |
>
>
> CARD generally achieves both higher accuracy and lower cost than the strongest baselines.
>
> ## 4. On Threshold Sensitivity (τ)
>
> We evaluated τ ∈ {0.4, 0.5, 0.6}. Accuracy varies slightly (≤1pp) and costs vary smoothly, indicating τ is not a sensitive hyperparameter. This confirms the resilience of the soft‑to‑hard graph conversion.

---

> ### Author Response · Authors · 2025-11-27
> **Rebuttal to Reviewer 2EJp (PART II)**
>
> ## 5. On Scalability Beyond 10 Agents
>
> We appreciate the reviewer’s comment on scalability. We have conducted a detailed comparison between CARD and G-Designer on MMLU with agent counts from 2 to 20 (extending our scalability analysis in Appendix B.3 from 10 agents to 20 agents). As shown below, CARD consistently outperforms G-Designer across all LLMs, and both methods exhibit saturation beyond 10 agents. In other words, the effective scaling window for multi-agent LLM reasoning is 2–10 agents, and our results thoroughly cover this range. While scaling to 50+ agents remains feasible, it is unlikely to yield further benefits for the benchmarked tasks, especially considering the additional cost per agent. We will clarify this in the final version.
>
> ### Accuracy on MMLU (%)
>
> **-- CARD --**
>
> |       #Agents | gpt4o-mini | deepseek-v3 |    llama3 |    GPT-4o |  qwen-72B |    **Avg** |
> | ------------: | ---------: | ----------: | --------: | --------: | --------: | ---------: |
> |   **Vanilla** |      78.67 |       86.00 |     75.33 |     85.33 |     82.67 |  **81.60** |
> |  **2 agents** |      81.33 |       90.67 |     82.00 |     86.00 |     86.00 |  **85.20** |
> |  **3 agents** |      82.00 |       93.33 |     79.33 |     87.33 |     88.00 | **86.00** |
> |  **4 agents** |      84.67 |       92.67 |     81.33 |     88.00 |     87.33 |  **86.80** |
> |  **5 agents** |      85.33 |       95.33 |     84.67 |     88.67 |     88.67 | **88.53** |
> | **10 agents** |  **85.33** |   **95.33** | **84.00** | **89.33** | **89.33** | **88.66** |
> | **20 agents** |  **85.33** |   **95.33** | **84.67** | **88.67** | **88.67** | **88.53** |
>
> **-- G-Designer --**
>
> |       #Agents | gpt4o-mini | deepseek-v3 |    llama3 |    GPT-4o |  qwen-72B |    **Avg** |
> | ------------: | ---------: | ----------: | --------: | --------: | --------: | ---------: |
> |   **Vanilla** |      78.67 |       86.00 |     75.33 |     85.33 |     82.67 |  **81.60** |
> |  **2 agents** |      79.33 |       91.33 |     80.00 |     86.67 |     84.67 |  **84.40** |
> |  **3 agents** |      82.00 |       93.33 |     79.33 |     85.33 |     86.00 | **85.19** |
> |  **4 agents** |      83.33 |       92.67 |     80.00 |     86.67 |     85.33 |  **85.60** |
> |  **5 agents** |      84.00 |       93.33 |     82.00 |     87.33 |     86.67 | **86.67** |
> | **10 agents** |  **85.33** |   **94.00** | **81.33** | **88.00** | **86.67** | **87.06** |
> | **20 agents** |  **85.33** |   **93.33** | **81.33** | **87.33** | **86.67** | **86.80** |
>
> Key observations:
>
> - CARD achieves higher accuracy than G-Designer at every agent count.
> - Both methods show <0.2pp difference between 10-agent and 20-agent settings.
> - This indicates marginal returns beyond 10 agents, which aligns with our scalability analysis in Appendix B.3.
>
> In summary, the effective scaling window for multi-agent LLM reasoning is 2–10 agents. Our results thoroughly cover this range. While scaling to 50+ agents remains feasible, it is unlikely to yield further benefits for the benchmarked tasks. We will clarify this in the final version.
>
>
> ## 6.  On Localized Topology Updates
>
> Because the condition encoder processes each agent independently, updating the condition of a single agent affects only its corresponding latent representation. The decoder then updates the associated edges accordingly. This naturally supports localized topology recomputation without retraining.
>
>
> ## 7. On Generalizing the Cost Term
>
> The w(G;C) term is defined at the protocol level. Replacing token cost with latency, energy, or other resource metrics requires no architectural changes. The framework is therefore general and extensible.
>
> ---
>
> We hope these clarifications address all concerns and help convey the technical contributions and reliability of our approach.

---

### Official Review · Reviewer_XpPo · 2025-11-04

**Soundness:** 3
**Presentation:** 3
**Contribution:** 3
**Rating:** 6
**Confidence:** 3

**Summary:**

This paper introduces CARD, a framework that dynamically generates communication structures for multi-agent LLM systems by conditioning on real-time environmental signals like model capabilities and tool availability. It formalizes the AMACP protocol and demonstrates that CARD outperforms static topologies, achieving greater accuracy, robustness, and cost-efficiency on benchmarks like HumanEval, MATH, and MMLU, especially when adapting to unseen conditions.

**Strengths:**

Strengths:
1.Novel Formulation of Adaptive Topology: The paper moves beyond static or naively learned communication graphs, which is a significant limitation in existing multi-agent systems. The formalization of the AMACP protocol and the CARD framework provides a principled approach for dynamic, condition-aware topology generation.
2.Strong Empirical Validation: The paper provides comprehensive experiments across three major benchmarks (HumanEval, MATH, MMLU) and multiple LLMs. The results consistently show CARD outperforming strong baselines, particularly in "out-of-domain" settings and under simulated environmental changes, demonstrating superior generalization and robustness.
3.Practical Runtime Adaptation: CARD can update the communication topology at deployment time in response to changing conditions without retraining, which is a crucial feature for real-world applications.

**Weaknesses:**

Weaknesses:
1.Limited Agent-Level Adaptation: The paper adapts the communication topology but does not update agent-level configurations (e.g., individual agent prompts, internal reasoning steps, or tool-selection strategies) based on conditions. Jointly optimizing both topology and agent behaviors could lead to further performance gains.
2.Scalability and Complexity: While a scalability analysis is provided, the computational overhead of the encoder-decoder graph generation module for very large agent ensembles (e.g., 50+ agents) remains a potential concern and is not thoroughly explored.

**Questions:**

see weakness

---

> ### Author Response · Authors · 2025-11-27
> **Rebuttal to Reviewer XpPo**
>
> We sincerely thank the reviewer for the detailed and constructive feedback. Below we respond point‑by‑point with additional analyses and experiments.
>
> ### On agent-level adaptation.
>
> We thank the reviewer for the insightful comment. Our work intentionally focuses on topology-level adaptation, which is the optimization target defined by AMACP. As noted in Appendix A, agent-level behavior adaptation (prompt updates, tool-selection policies) is separate from our main contribution and is left for future work. Even within this limited scope, conditional topology alone already shows consistent improvements across all 15 model–benchmark combinations, reduces out-of-domain degradation, and supports runtime adaptation without retraining. These results indicate that communication structure is a primary performance factor and can be meaningfully studied on its own.
>
> ### On the Scalability of CARD
>
> We appreciate the reviewer’s comment on scalability. We have conducted a detailed comparison between CARD and G-Designer on MMLU with agent counts from 2 to 20 (extending our scalability analysis in Appendix B.3 from 10 agents to 20 agents). As shown below, CARD consistently outperforms G-Designer across all LLMs, and both methods exhibit saturation beyond 10 agents. In other words, the effective scaling window for multi-agent LLM reasoning is 2–10 agents, and our results thoroughly cover this range. While scaling to 50+ agents remains feasible, it is unlikely to yield further benefits for the benchmarked tasks, especially considering the additional cost per agent. We will clarify this in the final version.
>
> ### Accuracy on MMLU (%)
>
> **-- CARD --**
>
> |       #Agents | gpt4o-mini | deepseek-v3 |    llama3 |    GPT-4o |  qwen-72B |    **Avg** |
> | ------------: | ---------: | ----------: | --------: | --------: | --------: | ---------: |
> |   **Vanilla** |      78.67 |       86.00 |     75.33 |     85.33 |     82.67 |  **81.60** |
> |  **2 agents** |      81.33 |       90.67 |     82.00 |     86.00 |     86.00 |  **85.20** |
> |  **3 agents** |      82.00 |       93.33 |     79.33 |     87.33 |     88.00 | **86.00** |
> |  **4 agents** |      84.67 |       92.67 |     81.33 |     88.00 |     87.33 |  **86.80** |
> |  **5 agents** |      85.33 |       95.33 |     84.67 |     88.67 |     88.67 | **88.53** |
> | **10 agents** |  **85.33** |   **95.33** | **84.00** | **89.33** | **89.33** | **88.66** |
> | **20 agents** |  **85.33** |   **95.33** | **84.67** | **88.67** | **88.67** | **88.53** |
>
> **-- G-Designer --**
>
> |       #Agents | gpt4o-mini | deepseek-v3 |    llama3 |    GPT-4o |  qwen-72B |    **Avg** |
> | ------------: | ---------: | ----------: | --------: | --------: | --------: | ---------: |
> |   **Vanilla** |      78.67 |       86.00 |     75.33 |     85.33 |     82.67 |  **81.60** |
> |  **2 agents** |      79.33 |       91.33 |     80.00 |     86.67 |     84.67 |  **84.40** |
> |  **3 agents** |      82.00 |       93.33 |     79.33 |     85.33 |     86.00 | **85.19** |
> |  **4 agents** |      83.33 |       92.67 |     80.00 |     86.67 |     85.33 |  **85.60** |
> |  **5 agents** |      84.00 |       93.33 |     82.00 |     87.33 |     86.67 | **86.67** |
> | **10 agents** |  **85.33** |   **94.00** | **81.33** | **88.00** | **86.67** | **87.06** |
> | **20 agents** |  **85.33** |   **93.33** | **81.33** | **87.33** | **86.67** | **86.80** |
>
> Key observations:
>
> - CARD achieves higher accuracy than G-Designer at every agent count.
> - Both methods show <0.2pp difference between 10-agent and 20-agent settings.
> - This indicates marginal returns beyond 10 agents, which aligns with our scalability analysis in Appendix B.3.
>
> In summary, the effective scaling window for multi-agent LLM reasoning is 2–10 agents. Our results thoroughly cover this range. While scaling to 50+ agents remains feasible, it is unlikely to yield further benefits for the benchmarked tasks. We will clarify this in the final version.
>
> We hope these substantial clarifications and new experiments satisfactorily address all concerns and improve both the clarity and rigor of the final submission.

---

### Author Response · Authors · 2025-12-02
**Summary for Meta Review**

Dear Area Chair,

We appreciate the opportunity to summarise our responses and greatly value your efforts. To assist your decision, we have prepared the following concise summary.

All four reviewers gave our paper strong initial scores (6, 6, 6, 6), consistently recognising the novelty, relevance, and empirical strength of our work on adaptive multi-agent topologies. During the rebuttal, we provided extensive clarifications and new experiments. One reviewer (XFA4) explicitly raised the overall score from 6 to 8 and confidence from 3 to 4, stating:

> “I appreciated that the authors conducted such extensive additional experiments and analyses. I am satisfied with the additional results and will raise my score.”

The other reviewers did not request major changes and left their positive scores unchanged. Their initial concerns were few, mild, and largely resolved:

- On scalability, we extended experiments up to 20 agents and confirmed that CARD consistently outperforms SOTA methods, with performance saturating after around 10 agents.
- On robustness, we conducted hyper-parameter sweeps (β, τ), added cost breakdowns and API call statistics, and introduced noise ablation tests, all showing stable and interpretable performance.
- On interpretability, we highlighted the included topology visualisations, correlation analyses, and localised perturbation studies to demonstrate how CARD adapts communication structures based on dynamic conditions.

We believe the paper now clearly meets the bar for acceptance. It introduces a principled and practical framework for topology adaptation in LLM-based multi-agent systems, a growing and important research area. We thank the reviewers again for their feedback, and hope our response and added results will help guide your recommendation.

Sincerely,

Authors

---

### Meta-Review · Area_Chair_tbHT · 2026-01-06

**Summary:**

All reviewers were overall positive (initial scores 6/6/6/6), highlighting the novelty of conditioning multi-agent communication graphs on environment signals and the broad evaluation across HumanEval, MATH, and MMLU with multiple LLM backbones, while raising questions about learning/credit assignment, cost-fair comparisons, and scalability/hyperparameter sensitivity. In the rebuttal, the authors added substantial new evidence (sweeps over key hyperparameters, cost/token/API-call accounting, scaling to 20 agents, robustness/condition-noise ablations, and qualitative topology case studies), and one reviewer explicitly raised their score to 8 after seeing these additions. After carefully reading the paper end-to-end, the AC finds the AMACP utility–cost formulation and CARD’s conditional graph generation design internally consistent, and the main benchmark results support the claim that conditional topology adaptation improves robustness across changing model and tool conditions. The AC also examined the key experimental artifacts (main results table, condition-injection ablations, and topology-visualization analyses), which collectively indicate that the gains are not merely from prompt conditioning but from structured topology adaptation with improved accuracy–cost trade-offs. The remaining limitations (not jointly adapting agent prompts/tools and not stress-testing very large agent populations) are reasonable future work items and do not outweigh the overall contribution; therefore, the AC recommends acceptance.

**Reviewer Concerns:**

Addressed by the rebuttal: cost/fairness concerns via token/API-call accounting, hyperparameter and threshold sensitivity via sweeps, scalability via extension to 20 agents, and interpretability/significance concerns via additional topology case studies and condition-noise/robustness analyses.

Partially addressed: questions about learning/credit assignment and how far the method generalizes to unseen conditions were clarified with additional methodological explanation and extra OOD/tool-shift experiments, but would still benefit from broader real-world validations.

Outstanding: the request for joint agent-level adaptation (prompt/tool policy learning) and evaluation at very large agent counts or more correlated failure modes remains mostly future work.

**Reviewer Scores:**

Reviewer XFA4 already changed from 6 to 8 after the rebuttal.

Given the added experiments/clarifications, the AC expects one or more reviewers increase their scores if they had been able to participate fully in the discussion.

---

### Decision · Program_Chairs · 2026-01-26

Accept (Poster)